# Improving cascade outcomes for active TB: A global systematic review and meta-analysis of TB interventions

Gifty Marley[1,2°], Xia Zou[3°], Juan Nie[4°], Weibin Cheng[5,6°], Yewei Xie[2], Huipeng Liao[2], Yehua Wang[2], Yusha Tao[2], Joseph D. Tucker[2,7], Sean Sylvia[2,8], Roger Chou[9], Dan Wu[2,7], Jason Ong[10], Weiming Tang[1,2]*

1 Dermatology Hospital of Southern Medical University, Guangzhou, China, 2 University of North Carolina Project-China, Guangzhou, China, 3 Global Health Research Center, Guangdong Provincial People's Hospital (Guangdong Academy of Medical Sciences), Southern Medical University, Guangzhou, China, 4 Department of Research and Education, Guangzhou Concord Cancer Center, Guangzhou, China, 5 Institute for Healthcare Artificial Intelligence Application, Guangdong Second Provincial General Hospital, Guangzhou, China, 6 School of Data Science, City University of Hong Kong, Hong Kong, China, 7 Faculty of Infectious and Tropical Diseases, London School of Health and Tropical Medicine, London, United Kingdom, 8 Department of Health Policy and Management, Gillings School of Global Public Health, University of North Carolina, Chapel Hill, North Carolina, United States of America, 9 Oregon Health & Science University, Portland, Oregon, United States of America, 10 School of Public Health and Preventive Medicine, Monash University, Melbourne, Australia

☯ These authors contributed equally to this work.
* weimingtangscience@gmail.com

**Data Availability Statement:** All relevant data are within the manuscript and its Supporting Information files.

## Abstract

### Background

To inform policy and implementation that can enhance prevention and improve tuberculosis (TB) care cascade outcomes, this review aimed to summarize the impact of various interventions on care cascade outcomes for active TB.

### Methods and findings

In this systematic review and meta-analysis, we retrieved English articles with comparator arms (like randomized controlled trials (RCTs) and before and after intervention studies) that evaluated TB interventions published from January 1970 to September 30, 2022, from Embase, CINAHL, PubMed, and the Cochrane library. Commentaries, qualitative studies, conference abstracts, studies without standard of care comparator arms, and studies that did not report quantitative results for TB care cascade outcomes were excluded. Data from studies with similar comparator arms were pooled in a random effects model, and outcomes were reported as odds ratio (OR) with 95% confidence interval (CI) and number of studies (k). The quality of evidence was appraised using GRADE, and the study was registered on PROSPERO (CRD42018103331). Of 21,548 deduplicated studies, 144 eligible studies were included. Of 144 studies, 128 were from low/middle-income countries, 84 were RCTs, and 25 integrated TB and HIV care. Counselling and education was significantly associated with testing (OR = 8.82, 95% CI:1.71 to 45.43; $I^2$ = 99.9%, k = 7), diagnosis (OR = 1.44, 95% CI:1.08 to 1.92; $I^2$ = 97.6%, k = 9), linkage to care (OR = 3.10, 95% CI = 1.97 to 4.86; $I^2$ =

**Funding:** This work was supported by the National Institute of Health (R34MH119963 to WT), the Key Technologies Research and Development Program (2022YFC2304900-4 to WT), National Nature Science Foundation of China (81903371 to WT), and CRDF Global (G-202104-67775 to WT). The funders had no role in study design, data collection and analysis, decision to publish, or preparation of the manuscript.

**Competing interests:** The authors have declared that no competing interests exist.

**Abbreviations:** CI, confidence interval; DOTS, directly observed therapy short-course; EPHPP, Effective Public Health Practice Project; GRADE, Grading Recommendations, Assessment, Development, and Evaluation; HICs, high-income countries; HIV, human immunodeficiency virus; LMICs, lower/middle-income countries; MDR, multidrug-resistant TB; NAAT, nucleic acid amplification test; OR, odds ratio; PLWH, persons living with HIV; RCT, randomized controlled trial; TB, tuberculosis; WHO, World Health Organization.

0%, k = 1), cure (OR = 2.08, 95% CI:1.11 to 3.88; $I^2$ = 76.7%, k = 4), treatment completion (OR = 1.48, 95% CI: 1.07 to 2.03; $I^2$ = 73.1%, k = 8), and treatment success (OR = 3.24, 95% CI: 1.88 to 5.55; $I^2$ = 75.9%, k = 5) outcomes compared to standard-of-care. Incentives, multisector collaborations, and community-based interventions were associated with at least three TB care cascade outcomes; digital interventions and mixed interventions were associated with an increased likelihood of two cascade outcomes each. These findings remained salient when studies were limited to RCTs only. Also, our study does not cover the entire care cascade as we did not measure gaps in pre-testing, pretreatment, and post-treatment outcomes (like loss to follow-up and TB recurrence).

## Conclusions

Among TB interventions, education and counseling, incentives, community-based interventions, and mixed interventions were associated with multiple active TB care cascade outcomes. However, cost-effectiveness and local-setting contexts should be considered when choosing such strategies due to their high heterogeneity.

## Author summary

### Why was this study done?

- Developing new and innovative interventions to improve tuberculosis (TB) care services use and successful treatment are essential to the global efforts to end TB.

- There is a limited scope on the overall impact of these interventions because most studies focus on interventions' capacity to enhance specific TB care outcomes.

- Evaluating existing evidence to ascertain the effect TB interventions on overall care cascade outcomes is paramount to informing holistic TB control strategies

### What did the researchers do and find?

- We systematically reviewed and meta-analyzed evidence on TB interventions and their effects on the TB care cascade for active TB from 144 peer-reviewed studies.

- In this study, the 5 out of 12 identified TB interventions associated with multiple care cascade outcomes were education and counseling, incentives, digital interventions, community-based, multisector collaborations, and mixed interventions.

- Among LMIC studies, education and counseling, incentives, community-based interventions, and multisector collaborations were the interventions associated with at least three TB care cascade outcomes.

### What do these findings mean?

- A wide range of relatively simple interventions could substantially improve TB care outcomes.

- Multistep efficient interventions like education and counseling, incentives, and mixed interventions should be keenly considered in expanding active TB control programs.

- Researchers should revise multistage effective interventions to incorporate local context needs due to their high heterogeneity.

## Introduction

Tuberculosis (TB) affected an estimated 10.6 million people and caused 1.6 million deaths in 2021 [1]. The United Nations Sustainable Development Goals and World Health Organization (WHO)'s End TB Strategy set ambitious global targets for significant reductions in the global TB burden by 2030 [2]. Summarizing the existing evidence is essential in planning future TB control programs, as additional efforts are strongly needed to attain the goal.

The TB care cascade comprises six fundamental steps: testing, diagnosis, linkage-to-care, cure, treatment completion, and treatment success [3,4]. Programmatic intervention refers to any public health intervention that seeks to prevent, promote health, or reduce the TB disease burden within a given population [5]. Many interventions like public education, staff training, mobile testing, and point-of-care testing have proven effective in enhancing TB services across the care cascade [6,7]. However, most intervention evaluations have focused on single TB care cascade outcomes, despite some affecting multiple care cascade outcomes.

Moreover, previous reviews have mainly focused on synthesizing evidence of interventions on single care cascade outcomes—per our knowledge [8,9]. This limited scope is likely due to most studies focusing on interventions' capacity to enhance specific care outcomes [10–13]. Recent studies have sought to assess intervention effects on multiple care cascade outcomes [14–16]. Yet, no current review has assessed the impacts of interventions across the whole TB care cascade. Evaluating existing evidence to ascertain the multistep effects capacity of TB care interventions across the care cascade is paramount to inform holistic prevention and control strategies for achieving the global End TB targets.

This global systematic review and meta-analysis aimed to synthesize evidence on TB interventions and their effects on the TB care cascade for active TB.

## Method

The study protocol was registered in PROSPERO (registration number CRD42018103331) (Protocol A in S1 File), and our report writing followed the PRISMA checklist [17].

### Search strategy and selection criteria

Four databases, including PubMed, Embase, CINAHL, and Cochrane trials registry, were searched using free text and controlled vocabulary terms (MeSH) for studies published from January 1970 till September 2022. The PICO framework informed search terms (Table 1).

The final PubMed search strategy included the following: ("tuberculosis, meningeal"[MeSH Terms] OR ("Tuberculosis"[Text Word] OR "TB"[Text Word])) AND 1848/01/01:2022/12/31 [Date—Publication] AND ("Uptake"[Title/Abstract] OR "Adherence"[Title/Abstract] OR "adhere"[Title/Abstract] OR "Compliance"[Title/Abstract] OR "comply"[Title/Abstract] OR "compliant"[Title/Abstract] OR "retain"[Title/Abstract] OR "retained"[Title/Abstract] OR "Retention"[Title/Abstract] OR "outcome"[Title/Abstract] OR "outcomes"[Title/Abstract] OR

**Table 1. Detail of PICO components that informed search strategy.**

| PICO | |
|------|---|
| P | Individuals living with TB (diagnosed or undiagnosed) or providers caring for these patients |
| I | Operational interventions delivered in conjunction with testing, care, or treatment of TB infection |
| C | Standard-of-care or no intervention |
| O | TB care cascade outcomes (testing, diagnosis, linkage-to-care and treatment outcomes) |

*Population*: Tuberculosis OR TB

*Intervention*: Intervention; Counseling; Education OR educate; Teach; Training; Program; Engagement; Smoking AND reduce, reduction, cessation

*Outcome*: Test OR tested OR testing; Diagnosis OR diagnostics OR diagnosed; Linking OR linkage OR linkage-to-care; Uptake; Retain OR retained OR retention; Adherence OR adhere; compliance OR comply.

"Testing"[Title/Abstract] OR "Diagnosis"[Title/Abstract] OR "Diagnostics"[Title/Abstract] OR "linkage-to-care"[Title/Abstract] OR "linkage-to-care"[Title/Abstract]) AND (("intervention"[Title/Abstract] OR "interventions"[Title/Abstract] OR "interventional"[Title/Abstract] OR "cohort*"[Title/Abstract] OR "trial"[Title/Abstract] OR "trials"[Title/Abstract] OR "RCT"[Title/Abstract]).

We also searched through the reference lists of similar published systematic reviews to identify studies not captured by our database search outcomes. Details of search outcomes are in Table B in S1 File.

## Eligibility

Peer-reviewed articles, abstracts, and clinical trials that met the search criteria were screened for eligibility. Randomized control trials (RCTs) and non-RCTs with comparator arms that implemented nonpharmaceutical interventions were eligible. Studies that reported outcomes on at least one care cascade outcome were eligible (Fig 1). Studies reporting the use of pharmaceutical interventions (studies that reported change in TB drug regimens or introduced new lines of TB drugs as interventions) were excluded, as recent reviews have evaluated DOTS efficiency in improving TB care outcomes [8,18]. We also excluded dissertations, systematic reviews, studies on latent TB, qualitative studies, mathematical modeling/simulation studies, quantitative studies without a standard-of-care control group, studies that did not report quantitative data on care cascade outcomes, and short reports.

## Screening

WT and YH independently screened the title and abstracts for eligibility using Covidence. YW and YS, JN and YS, and GM and CH screened the eligible full texts for inclusion, and WT resolved full-text discrepancies.

## Data extraction and quality assessment

WC and WT and HL and JN double-extracted data from included studies using a designed spreadsheet, and XZ resolved discrepancies. Data extracted included first author, publication year, country, target population, study settings, designs, type of interventions, TB care cascade outcomes, and sample size. We assessed the risk of bias using the Cochrane Risk of Bias tool and the quality of observational studies with the Effective Public Health Practice Project (EPHPP) Quality Assessment Tool [19–21]. The EPHPP tool assessed each study in seven main domains (selection bias, study design, confounders, blinding, data collection, methods,

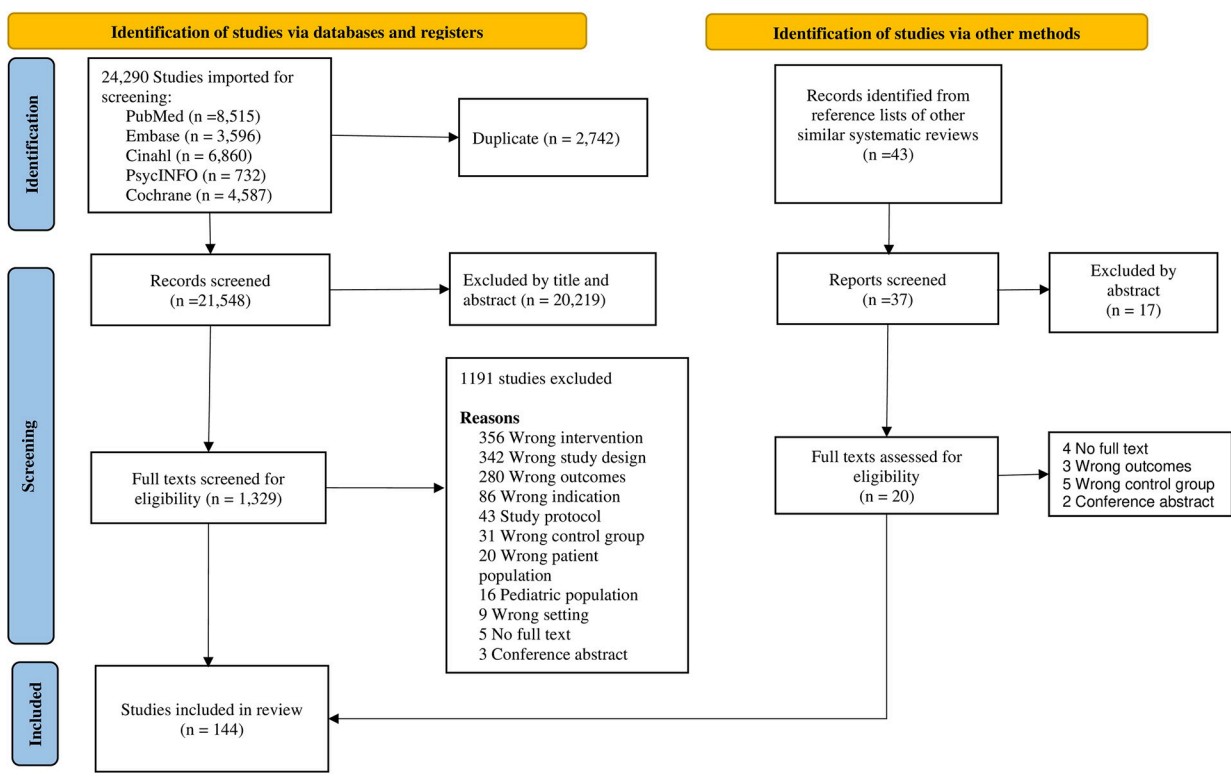

**Fig 1. Adapted PRISMA flowchart.**

and withdrawals and dropouts of patients) and rated studies as strong, moderate, or weak quality.

## Definitions

TB testing in this review refers to the initial screening tests that are administered to persons suspected of TB after evaluation of risk and symptoms [22]. It aims to sort persons who probably have a disease from those who do not and is not intended to be a diagnostic test. Screening tests use a simpler testing process (usually a sputum smear test for bacteriologically testing, chest X-ray for pulmonary TB and low complexity nucleic acid amplification tests (NAATs) for simultaneous initial resistance testing) [23,24], after which persons with positive or clinically suspicious results (like inconclusive results) are referred for diagnostic testing [25]. Diagnosis refers to administering comprehensive clinical evaluation and laboratory tests to confirm a screen-tested person as having TB by using at least one approved diagnostic test approach [26]. Diagnostic tests for active TB include Xpert MTB/RIF (for multidrug-resistant and rifampicin-resistant TB), cartridge-based NAAT methods (like TrueNat), biopsy tissue culture (for extrapulmonary TB), Xpert Ultra assay, and X-ray [25]. Linkage-to-care in this review refers to the stage in care starting from registering newly diagnosed TB patients at designated facilities to initiate treatment or successfully initiating TB patients on treatment. Treatment completion refers to TB patients who finished the required treatment course but without evidence of failure or cure [27]. Cured was defined as patients' bacteriologically confirmed TB positive at the beginning of treatment but with a smear- or culture-negative result in the last month of treatment and three or more consecutive cultures taken at least 30 days apart are negative after the intensive phase [27]. Treatment success describes the total number of persons diagnosed with TB who completed treatment and were cured [27].

Mixed interventions refer to strategies that merge or concurrently implement two or more interventions. For example, adopting a digital intervention and using incentives to facilitate linkage-to-care (Table 2).

**Table 2. Definitions of care cascade outcomes, interventions, and strategies discussed in this review.**

| Variables | Definitions |
|---|---|
| Intervention | Any initiative implemented to improve TB care outcomes. This includes the following:<br> - policy introduction or change (like updating clinical guidelines, integrating TB care with other services, establishing new referral procedures, and fostering collaboration between departments)<br> - provision of tools and resources to improve case detection and treatment outcomes (like providing X-ray machines, microscopes, GeneXpert machines, pill boxes, etc.)<br> - capacity building (like staff training, providing educational materials for TB patients, and engaging lay workers in services delivery) |
| TB Testing | Persons deemed at risk of TB who have received a screening test (like sputum smear test) to check if they have been infected with TB bacteria. Screening tests are not diagnostic tests and persons with positive or clinically suspicious results (like inconclusive results) are referred for diagnostic testing to confirm diagnosis. |
| TB Diagnosis | Persons diagnosed with any form of TB through clinical evaluation using at least one diagnostic testing method. Some examples of diagnostic tests for active TB include Xpert MTB/RIF (for multidrug-resistant and rifampicin-resistant TB), cartridge-based NAAT methods (like TrueNat), biopsy tissue culture (for extrapulmonary TB), Xpert Ultra assay, and X-ray |
| Linkage-to-care | Diagnosed persons living with TB who are successfully registered to initiate or have initiated patient-centered treatment (including directly observed therapy short-course (DOTS) treatment at designated health facilities within any time after diagnosis). |
| Treatment completed | Persons diagnosed with TB who finish the required treatment course without evidence of failure or cure. |
| Cured | Persons bacteriologically confirmed TB positive at the beginning of treatment with a smear- or culture-negative results in the last month of treatment and three or more consecutive cultures taken at least 30 days apart are negative after the intensive phase. |
| Treatment success | Treatment success refers to diagnosed TB patients who complete their treatment regimen and are cured |
| Interventions | |
| Mixed interventions | A comprehensive intervention consisting of two or more strategies implemented concurrently or merged to form tailored strategies. Mixed interventions identified in this review include the following:<br> - staff training patient education (clinical staff received training to actively educate and screen patients for TB. They also received informational materials for distribution to patients as part of patient education)<br> - active case finding, and education and counseling (clinical staff and community health workers embark on active case finding outreach in high-risk communities, community volunteers helped educate the community members on the risks of TB and the need to get tested, and newly diagnosed TB patients received counseling on TB treatment, managing adverse effects and the importance of adherence)<br> - Onsite sputum collection, expediated diagnosis and treatment initiation, patient education and counseling<br> - Staff training, revised guidelines to improve facility-based patient care |
| Staff training | Ad hoc or routine training for healthcare workers or engaged community lay workers on TB services delivery. This intervention strategies included the following:<br> - Training nurses and clinicians on TB, TB treatment, diagnosis, patient-centered therapy, and adherence counseling<br> - Training clinic staff of alcohol use evaluation, patient counseling the need for tobacco use cessation, referral systems and active case finding through patient screening<br> - Introducing clinic staff to new facility-based TB care procedures, and digital systems to aid treatment observation |

*(Continued)*

**Table 2.** (Continued)

| Variables | Definitions |
|---|---|
| Active case finding | Interventions that encourage high-at-risk persons to present to designated sites for TB testing. Active case finding methods identified in this review include the following:<br>  - engaging community health workers/nurses/physicians and volunteer peer educators to introduce and assist in community TB screening testing<br>  - initiating household testing and contact tracing for all newly diagnosed TB patients and for individuals at high risk of TB infection.<br>  - Training healthcare staff of other clinics and health departments to actively screen patients for TB.<br>  - Training community-based pharmacists to actively screen suspected clients for TB and refer clients with positive results. |
| Education and counseling | Impacting knowledge about TB or self-care TB to patients through information materials dissemination, one-on-one talk sessions, and public education (like peer education and in-school talk sessions). Most studies used the term "counseling" loosely to describe one-on-one sessions between designated counsellors and TB patients about TB treatment, the need for adherence and coping with adverse effects. Types of education and counseling interventions identified through this review included the following:<br>  - *support groups* (TB patients form/join groups to support each other psychologically and emotionally through the treatment journey)<br>  - *psychosocial education and counseling* (trained counsellors counsel newly diagnosed TB patients and patients on treatment on TB infection and treatment)<br>  - *lay counseling* (community lay workers engaged to provide home-based care and adherence support for TB patients)<br>  - *public education* (healthcare workers or lay counselors engage communities and household member of newly diagnosed TB patients/high-risk populations on TB infection, risks of transmission and the need to get tested<br>  - *practice-based staff education* (tuberculosis specialists visit intervention clinic sites to promote tuberculosis screening, raise awareness of TB as a local public health concern, and distributed copies of local tuberculosis screening guidelines among healthcare providers)<br>  - *peer education* (peer educator volunteers recruited and trained on TB to educate peers within specified geographical locations on TB transmission, TB risk groups, how treatment is conducted, the importance of screening). |
| Incentive | Offering compensation to patients to encourage TB services utilization. Types of incentives identified in this review included the following:<br>  - *financial incentives* (individuals received monetary incentives as transport reimbursement)<br>  - *nonfinancial incentives* (providing food and provision coupons, airtime cards, and so on as an incentive to promote patient return visits for TB test screening, test results, treatment initiation, or treatment refills). |
| Digital interventions | The use of digital appliances (like smartphones) and online applications (like social media) as tools to facilitate TB services delivery (like education and treatment observation). Digital approaches used in studies included in this review were as follows:<br>  - SMS reminders (care providers exchanged SMS with patients to confirm medication adherence. This could be staff sending reminder texts, or patients sending agreed upon texts to confirm they have taken the pills for the day)<br>  - Calls (treatment monitors call patients on phone or via agreed upon social media platforms to observe treatment adherence and provide treatment support if needed)<br>  - phone-based apps (developing an innovative mHealth tool that uses basic mobile phones to monitor and improve adherence to TB drugs)<br>  - digital diagnosis (like computer-aided chest X-ray interpretation) |
| Home-based Care | Community healthcare workers, nurses, and volunteer peers provide doorstep TB care at home to persons living with TB who for some reason cannot report to designated treatment sites. |

(*Continued*)

**Table 2.** (Continued)

| Variables | Definitions |
|---|---|
| Tobacco and alcohol use control | Implementing interventions to reduce tobacco use or alcohol use among a target population. Types of tobacco and alcohol control interventions identified include the following:<br>  - enrolling TB patients into tobacco cessation programs (nicotine replacement therapy)<br>  - orally administered naltrexone for patients with opioid abuse problem for 6 months, usually within 2 weeks of TB treatment initiation.<br>  - Training of family members and care providers of TB patients on how to assess alcohol use and developing a structured intervention manual and visual aids to explain TB, the effect of alcohol on the human body, loved ones, and on TB, alcohol being a risk factor for TB, and effects of alcohol on treatment adherence.<br>  - Continuous patient counseling on alcohol use reduction |
| Community-based intervention | Interventions implemented at the community level outside of an established standard facility. Identified community-based interventions included the following:<br>  - point-of-care testing and treatment; one-stop shops for integrated services delivery<br>  - engagement of community nurses, lay workers, and peer educators<br>  - mobile X-ray and GeneXpert testing<br>  - field sputum collection; marketplace/school/workplace TB screening, etc.<br>  - community care workers (CCWs), providing stipend for CCWs and enhanced supervision of CCWs to provide comprehensive TB care |
| Multisector collaborations | Partnerships between two or more healthcare departments or nonclinical institutions to improve TB services delivery and patient care. For example: (1) referral systems with community-based pharmacies to facilitate linkage-to-care; (2) integration of HIV and TB care at in ART centers to oversee TB treatment in coinfected PLWH; (3) formation of consortium between private clinics, local pharmacies, and public clinics to facilitate TB referrals, integrating TB services in chest clinics, and so on. |

Definitions adopted from references [26,27].

## Data analysis

We reviewed the outcomes individually and included studies that reported mixed interventions in the data synthesis and meta-analyses.

## Data pooling

Data from RCTs and non-RCTs with similar comparator arms and intervention strategies compared to standard-of-care were pooled in a meta-analysis using the Review Manager (The Cochrane Collaboration, 2014; Version 5.3). We utilized a random-effect model in pooling the data, funnel plots to assess publication bias and Egger's test to assess small sample size effects as a potential marker of publication bias ($p < 0.05$). The results are reported as odds ratios (OR) with corresponding 95% confidence intervals (CI), the number of studies (k), heterogeneity ($I^2$), and certainty of evidence quality.

## Subgroup analysis and risk assessment

Causes of heterogeneity were exploited in subgroup analyses stratified by country designation according to the 2020 World Bank ranking (LMICs versus HICs), study designs (RCTs versus non-RCTs), and HIV services integration. The certainty of the evidence quality for each outcome was appraised using the Cochrane Grading Recommendations, Assessment, Development, and Evaluation (GRADE).

## Quality assessment

In an analysis of quality assessment, studies were stratified based on study design and level of evidence. Bias among randomized controlled studies was assessed using the Cochrane

Collaboration "Risk of Bias" tool, using six criteria in four sources of bias: selection bias, performance and detection bias, attrition bias, and reporting bias. Bias in other quantitative studies was assessed using the Newcastle–Ottawa Quality Assessment (EPHPP) Scale, which assessed selection bias, patient-level barrier, and measurement bias. The EPHPP tool assessed each study in seven main domains (selection bias, study design, confounders, blinding, data collection, methods, and withdrawals and dropouts of patients) and rated each aspect as strong, moderate, or weak quality. Results of the quality assessment were used in estimating the quality of evidence as part of the GRADE assessment for each intervention. The quality of evidence was assessed according to the methodology described by the GRADE working group. A GRADE table was generated for each meta-analysis outcome and sub-analysis.

This study is reported as per the Preferred Reporting Items for Systematic Reviews and Meta-Analyses (PRISMA) guideline (S1 PRISMA Checklist in S1 File).

## Results

We identified 21,548 deduplicated articles from the database search and 36 studies from searching reference lists of similar systematic reviews, and finally, 144 studies were included in this review (Fig 1). Thirty-two studies (64.4%) were published before 2010, and 25 (14.8%) targeted persons living with HIV (PLWHs). There were 84 (58.3%) RCTs and 31 (21.5%) observational, and 126 (87.5%) from LMICs (Table C in S1 File). By care cascade outcomes, 92 (63.9%) studies reported intervention effects on single care cascade outcomes, and 26 (20.1%) studies appraised the effects of more than one intervention strategy. We identified a total of 12 major TB interventions across the six care cascade outcomes of interest (Fig 2). Among single interventions, most studies assessed incentives (11.8%) and digital interventions (11.8%), while staff training and multisector collaborations were the least appraised singular interventions (4.2%) each. See Table 3 for further details.

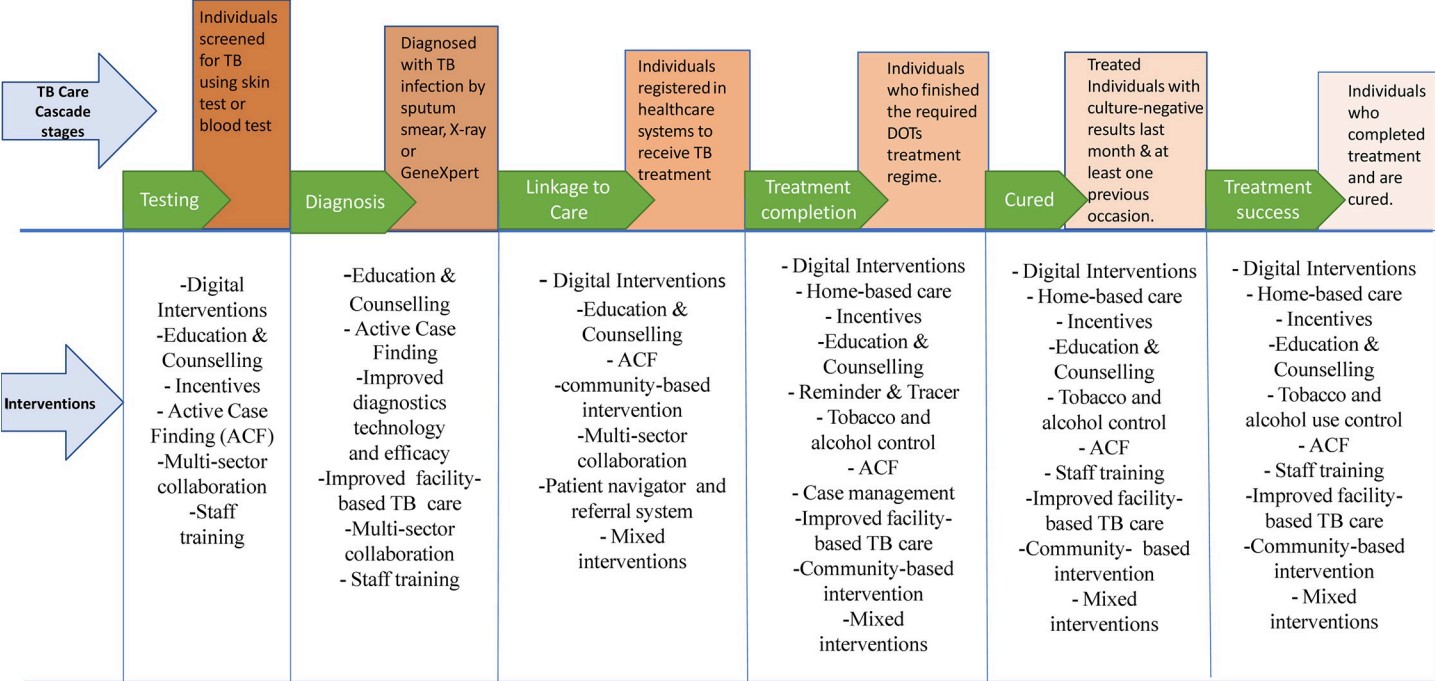

**Fig 2. Summary showing the various programmatic interventions associated with outcomes at each stage of the TB care cascade.**

**Table 3. Summary showing characteristics of all eligible studies included in this review (N = 144).**

| Characteristic | | Number of studies (%) |
|---|---|---|
| Year of publication | | |
| | Before 2010 | 32 (22.5) |
| | 2010–2022 | 114 (80.3) |
| Study design | | |
| | RCT | 84 (56.3) |
| | Non-RCT | 29 (20.4) |
| | Pre- and Post-intervention | 31 (21.8) |
| | Other observational studies | 2 (1.4) |
| Integration | | |
| | With HIV | 25 (17.6) |
| | Without HIV | 121 (25.2) |
| Regional settings | | |
| | LMICs | 126 (88.7) |
| | HICs | 18 (12.7) |
| Number of care cascade outcomes reported | | |
| | 1 | 92 (64.7) |
| | 2 | 16 (11.3) |
| | 3 | 27 (19.0) |
| | >3 | 9 (6.3) |
| Interventions identified | | |
| | Education and counseling only | 11 (7.7) |
| | Incentives only | 17 (12.0) |
| | Active case finding only | 15 (10.6) |
| | Multisector collaborations only | 6 (4.2) |
| | Community-based interventions only | 8 (5.6) |
| | Staff training | 6 (4.2) |
| | Digital interventions | 17 (12.0) |
| | Tracers and reminders | 7 (4.9) |
| | Mixed intervention | 26 (18.3) |
| | Other single interventions | 31 (21.8) |
| Studies by cascade outcome (*n* = 92) | | |
| | TB testing only | 11 (12.0) |
| | TB diagnosis only | 22 (23.9) |
| | Linkage to care only | 15 (16.3) |
| | Treatment completion only | 21 (22.8) |
| | Cured only | 9 (9.8) |
| | Treatment success | 14 (15.2) |
| Study populations | | |
| | TB patients (newly diagnosed and existing) | 72 (50.0) |
| | MDR and pulmonary TB patients | 16 (11.1) |
| | Persons living with HIV | 8 (5.6) |
| | Persons who use drugs | 3 (2.0) |
| Study settings | | |
| | Facilities | 98 (68.1) |
| | Community | 42 (29.2) |
| | Other settings | 3 (2.0) |

RCTs, randomized controlled trials; non-RCT, includes quasi-experimental trials; LMICs, lower/middle-income countries as designated by the World Bank in 2021; HICs, high-income countries; Facility, includes primary healthcare centers, laboratories, private hospitals, research centers, clinics, rehabilitation centers, and so on; MDR, multidrug-resistant TB.

On single outcomes by cascade, 11 (12.2%) studies reported TB testing outcomes [28–38], 22 (23.9%) reported TB diagnosis [39–57], 15 (16.3%) reported on linkage to care [58–74], 21 (22.8%) studies reported treatment completion [11,12,75–94], 9 (9.8%) reported on cure [14,80,95–102], and 14 (15.2%) studies reported on treatment success [80,103–116].

Among the 84 RCTs, 34 (36.9%) had a low risk of selection bias, 48 (41.7.0%) had a high risk of performance bias, 47 (56.0%) had a low risk of attrition bias, and 45 (53.6%) had an unclear risk of reporting bias (Table E in S1 File). Among the 60 non-RCT studies, 12 (20.0%) were rated high quality, 32 (53.3%) were moderate, and 16 (26.7%) had poor quality (Table F in S1 File). GRADE assessment score showed 19/54 outcomes (35.2%) were of high evidence certainty, 24 (44.4%) were moderate, eight (14.8%) were low, and four (7.1%) were very low (Table G in S1 File). Funnel plots and Egger's test suggested potential publication bias for studies reporting TB diagnosis (p_Egger = 0.029) and treatment completion (p_Egger = 0.0052). The plots were adjusted to correct for publication bias using and the trim and fill method (S1 Fig).

The interventions were highly heterogeneous ($I^2 > 50\%$) due to variations in components, intensity, training/resources required, and other factors. For example, mixed interventions varied in the number and types of interventions combined. Meta-regression results showed that study design, year of publication, and region of the study were the main sources of heterogeneity among interventions with effects on multiple TB care cascade outcomes. The main source of heterogeneity in mixed interventions reporting linkage to care was study design (OR = 9.44, 95% CI: 1.04 to 85.59, $p = 0.046$). Similarly, the source of heterogeneity in studies appraising the use of community interventions (OR = 1.46, 95% CI: 1.00 to 2.14, $p = 0.051$) and digital interventions (OR = 1.67, 95% CI: 1.11 to 2.51, $p = 0.013$) in TB cure was study design; and the source of heterogeneity in the use of incentives (OR = 1.03, 95% CI: 1.00 to 1.05, $p = 0.039$) and alcohol and tobacco use control (OR = 0.88, 95% CI: 0.81 to 0.96, $p < 0.01$) among studies reporting treatment success outcomes was year of publication. Additionally, the source of heterogeneity among studies assessing community-based and education and counseling interventions in treatment completion were year of publication (OR = 0.96, 95% CI: 0.93 to 0.99, $p = 0.014$) as well as study design (OR = 0.25, 95% CI: 0.12 to 0.52, $p < 0.01$) and region (OR = 0.76, 95% CI: 0.63 to 0.92, $p < 0.01$), respectively (Table D in S1 File).

## Care cascade outcomes

Among the interventions identified, seven interventions (education and counseling, incentives, digital interventions, community-based interventions, multisector collaborations, mixed interventions, and reminders and tracers) were significantly associated with outcomes at multiple stages of the care cascade (Table 4).

## TB testing

Among interventions affecting outcomes at multiple stages of the care cascade, education and counseling (OR = 8.82, 95% CI: 1.71 to 45.43; $I^2$ = 99.9%, k = 7; high certainty), incentives (OR = 1.74, 95% CI: 1.63 to 1.85]; $I^2$ = 37.2%, k = 5; low certainty), multisector collaborations (OR = 4.14, 95% CI: 3.42 to 5.01; $I^2$ = 99.2%, k = 2; low certainty), and digital interventions (OR = 1.97, 95% CI: 1.28 to 3.04; $I^2$ = 75.1%, k = 2; high certainty) were significantly associated with an increased likelihood of testing (Fig 3).

## TB diagnosis

Only education and counseling (OR = 1.44, 95% CI: 1.08 to 1.92; $I^2$ = 97.6, k = 9; high certainty) and multisector collaborations (OR = 8.00, 95% CI: 1.53 to 41.84; $I^2$ = 99.8, k = 5; low

**Table 4. Interventions significantly associated with TB care outcomes at multiple stages of the TB care cascade.**

| | TB testing | TB diagnosis | Linkage to care | Treatment completion | Cured | Treatment success |
|---|---|---|---|---|---|---|
| Intervention | OR [95% CI]; k | OR [95% CI]; k | OR [95% CI]; k | OR [95% CI]; k | OR [95% CI]; k | OR [95% CI]; k |
| Education and counseling | 8.82 [1.71–45.43]; k = 7 | 1.44 [1.08–1.92]; k = 9 | 3.10 [1.97–4.86]; k = 1 | 1.48 [1.07–2.03]; k = 8 | 2.08 [1.11–3.88]; k = 4 | 3.24 [1.88–5.55]; k = 5 |
| Incentives | 1.74 [1.63–1.85]; k = 5 | - | 2.86 [1.25–6.50]; k = 4 | 1.37 [1.10–1.71]; k = 12 | 1.62 [1.06–2.48]; k = 5 | 1.08 [1.05–1.11]; k = 5 |
| Community-based intervention | - | - | 9.91 [1.86–52.74]; k = 4 | - | 2.53 [1.92–3.35]; k = 9 | 2.91 [2.01–4.21]; k = 10 |
| Multisector collaboration | 4.14 [3.42–5.01]; k = 2 | 8.00 [1.53–41.84]; k = 5 | 3.25 [2.05–5.14]; k = 3 | - | - | - |
| Reminder and tracers | - | - | - | 1.03 [1.00–1.07]; k = 6 | - | 1.09 [1.01–1.16]; k = 2 |
| Digital interventions | 1.97 [1.28–3.04]; k = 2 | - | 1.10 [1.04–1.17]; k = 4 | - | - | - |
| Mixed interventions | - | - | - | - | 1.19 [1.13–1.26]; k = 3 | 1.14 [1.09–1.19]; k = 3 |

**OR**, odds ratio; **CI,** confidence interval; **k,** number of studies.

-: No study assessed how this intervention affected this outcome or the effects of this intervention on this outcome was not statistically significant.

certainty) were each associated with an increased likelihood of TB diagnosis among interventions that affected outcomes at multiple stages of the care cascade (Fig 4).

## Linkage-to-care

Among interventions that affected multiple care cascade outcomes, education and counseling (OR = 3.10, 95% CI: 1.97 to 4.86; $I^2 = 0$, k = 1; high certainty), incentives (OR = 2.86, 95% CI: 1.25 to 6.50; $I^2 = 86.2\%$ k = 4, moderate certainty), community-based interventions (OR = 9.91, 95% CI: 1.86 to 52.74; $I^2 = 99.6$, k = 4; moderate certainty), multisector collaborations (OR = 3.25, 95% CI: 2.05 to 5.14; $I^2 = 86.5$, k = 3; moderate certainty), and digital interventions (OR = 1.10, 95% CI: 1.04 to 1.17; $I^2 = 0.1\%$, k = 4; moderate certainty) were each associated with an increased likelihood of linkage-to-care (Fig 5).

| | | **TB testing** | | | | |
|---|---|---|---|---|---|---|
| Intervention | Intervention (Event/Total) | Control (Event/Total) | No. of studies | | Odds ratio (95% CI) | $I^2$ (%) |
| **Patients** | | | | | | |
| Digital interventions | 529/8859 | 291/8293 | (n=2) | | 1.97 (1.28, 3.04) | 75.12 |
| Incentives | 1101/1430 | 654/1475 | (n=5) | | 1.74 (1.63, 1.85) | 37.21 |
| Education and counseling | 20920/36335 | 2763/35033 | (n=7) | | 8.82 (1.71, 45.43) | 99.86 |
| **Providers** | | | | | | |
| Active case finding | 5047/283046 | 3651/185264 | (n=7) | | 1.68 (0.68, 4.13) | 99.80 |
| Multi-sector collaborations | 666/7086 | 261/8334 | (n=2) | | 4.14 (3.42, 5.01) | 99.18 |
| Staff training | 2637/200714 | 2658/209564 | (n=1) | | 1.04 (0.98, 1.09) | 0.00 |

0.0 2.0 4.0 6.0 8.0 10.0

**Fig 3. Forest plots showing the effects of various interventions on TB testing outcomes for active TB.**

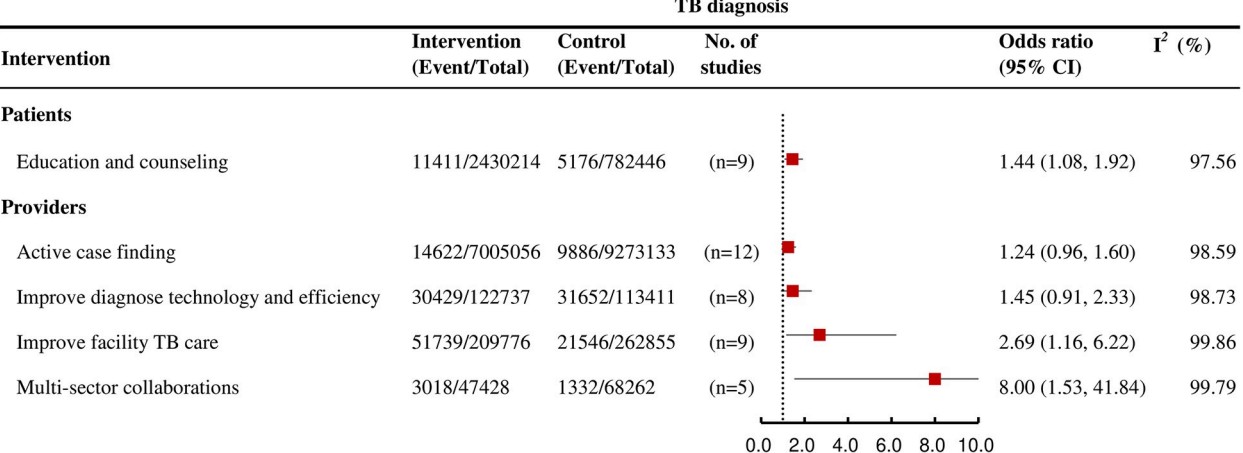

**Fig 4. Forest plots showing the effects of various interventions on TB diagnosis outcomes for active TB.**

## TB cure

Education and counseling (OR = 2.08, 95% CI: 1.11 to 3.88; $I^2$ = 76.7%, k = 4, moderate certainty), incentives (OR = 1.62, 95% CI: 1.06 to 2.48; $I^2$ = 98.5%, k = 5; high certainty), community-based interventions (OR = 2.53, 95% CI: 1.92 to 3.35; $I^2$ = 97.4%, k = 9; moderate certainty), and mixed interventions (OR = 1.19, 95% CI: 1.13 to 1.26; $I^2$ = 0% k = 3, high certainty) were each significantly associated with an increased likelihood of TB cure among the interventions that affected outcomes at multiple stages (Fig 6).

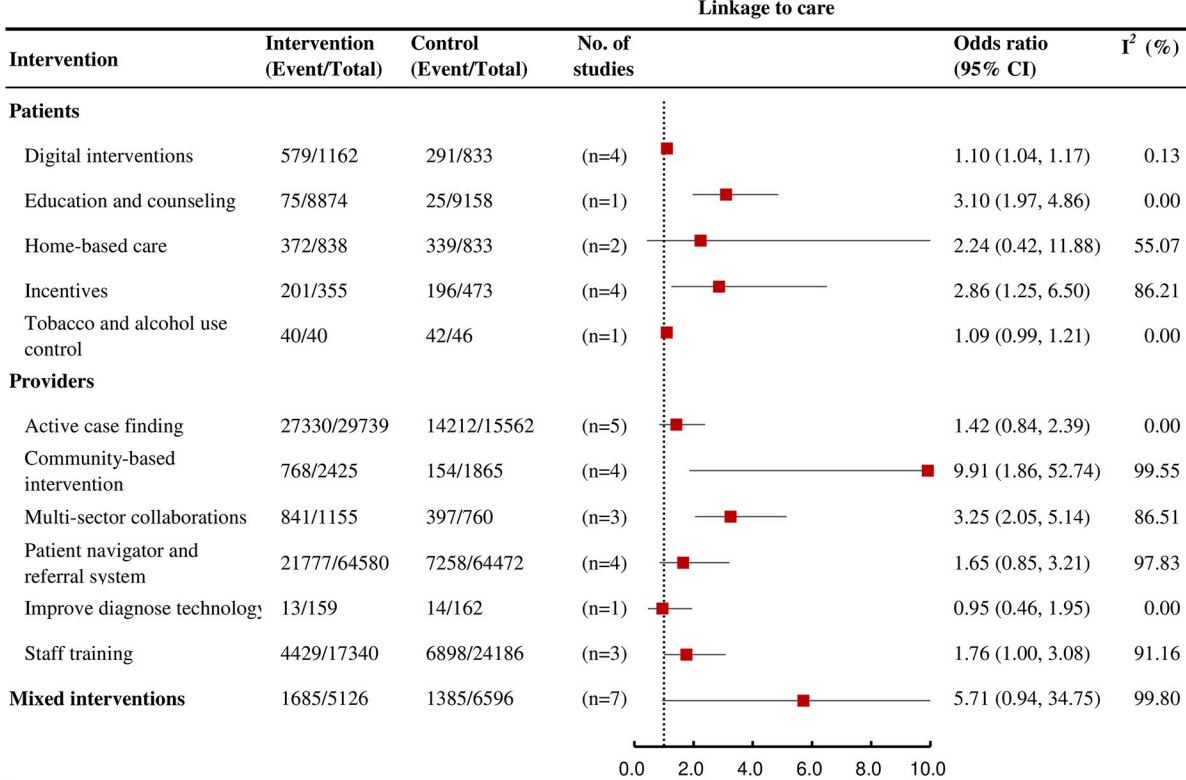

**Fig 5. Forest plots showing the effects of various interventions on TB linkage to care outcomes for active TB.**

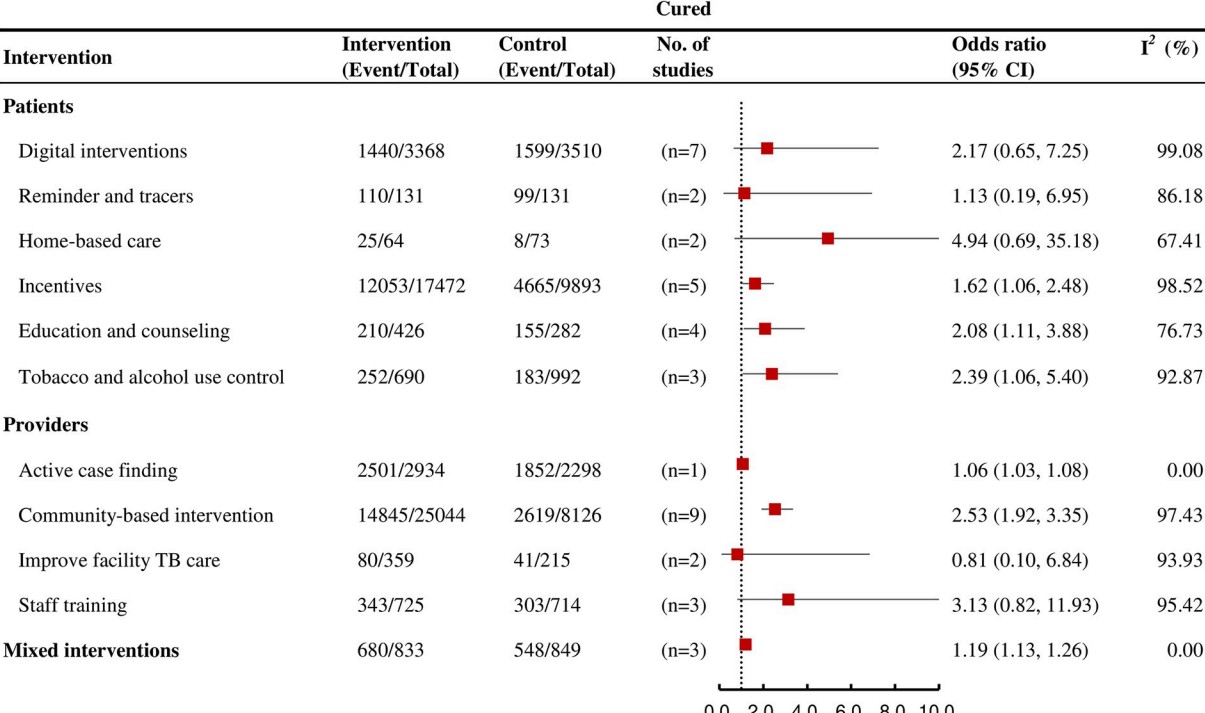

| Intervention | Cured | | | | Odds ratio (95% CI) | $I^2$ (%) |
|---|---|---|---|---|---|---|
| | Intervention (Event/Total) | Control (Event/Total) | No. of studies | | | |
| **Patients** | | | | | | |
| Digital interventions | 1440/3368 | 1599/3510 | (n=7) | | 2.17 (0.65, 7.25) | 99.08 |
| Reminder and tracers | 110/131 | 99/131 | (n=2) | | 1.13 (0.19, 6.95) | 86.18 |
| Home-based care | 25/64 | 8/73 | (n=2) | | 4.94 (0.69, 35.18) | 67.41 |
| Incentives | 12053/17472 | 4665/9893 | (n=5) | | 1.62 (1.06, 2.48) | 98.52 |
| Education and counseling | 210/426 | 155/282 | (n=4) | | 2.08 (1.11, 3.88) | 76.73 |
| Tobacco and alcohol use control | 252/690 | 183/992 | (n=3) | | 2.39 (1.06, 5.40) | 92.87 |
| **Providers** | | | | | | |
| Active case finding | 2501/2934 | 1852/2298 | (n=1) | | 1.06 (1.03, 1.08) | 0.00 |
| Community-based intervention | 14845/25044 | 2619/8126 | (n=9) | | 2.53 (1.92, 3.35) | 97.43 |
| Improve facility TB care | 80/359 | 41/215 | (n=2) | | 0.81 (0.10, 6.84) | 93.93 |
| Staff training | 343/725 | 303/714 | (n=3) | | 3.13 (0.82, 11.93) | 95.42 |
| **Mixed interventions** | 680/833 | 548/849 | (n=3) | | 1.19 (1.13, 1.26) | 0.00 |

**Fig 6. Forest plots showing the effects of various interventions on TB cure outcomes for active TB.**

## Treatment completion

Among interventions affecting multiple care cascade outcomes, education and counseling (OR = 1.48, 95% CI: 1.07 to 2.03; $I^2$ = 73.1%, k = 8; high certainty), incentives (OR = 1.37, 95% CI: 1.10 to 1.71; $I^2$ = 88.3%, k = 12; moderate certainty), and reminder and tracers (OR = 1.03, 95% CI: 1.00 to 1.07; $I^2$ = 15.2%, k = 6; high certainty) were each significantly associated with an increased likelihood of TB treatment completion (Fig 7).

## Treatment success

Among the interventions with multistage effects on care cascade outcomes, education and counseling (OR = 3.24, 95% CI: 1.88 to 5.55; $I^2$ = 75.9%; k = 5, moderate certainty), incentives (OR = 1.08, 95% CI: 1.05 to 1.11; $I^2$ = 0% k = 5; high certainty), community-based interventions (OR = 2.91, 95% CI: 2.01 to 4.21; $I^2$ = 97.6, k = 10; moderate certainty), reminders and tracers (OR = 1.09, 95% CI: 1.01 to 1.16; $I^2$ = 33.4, k = 2; moderate certainty), and mixed interventions (OR = 1.14, 95% CI: 1.09 to 1.19; $I^2$ = 44.7%, k = 3; moderate certainty) were each significantly associated with increased likelihood of treatment success (Fig 8).

Supplementary Table G shows the generated GRADE tables for each outcome of interest (Table G in S1 File).

## Subgroup analysis

**RCTs and non-RCTs.** Among RCT and non-RCT studies, seven interventions (education and counseling, incentives, community-based intervention, digital interventions, active case finding, mixed interventions, and multisector collaboration) were identified to be associated with multiple TB care cascade outcomes (Table 5).

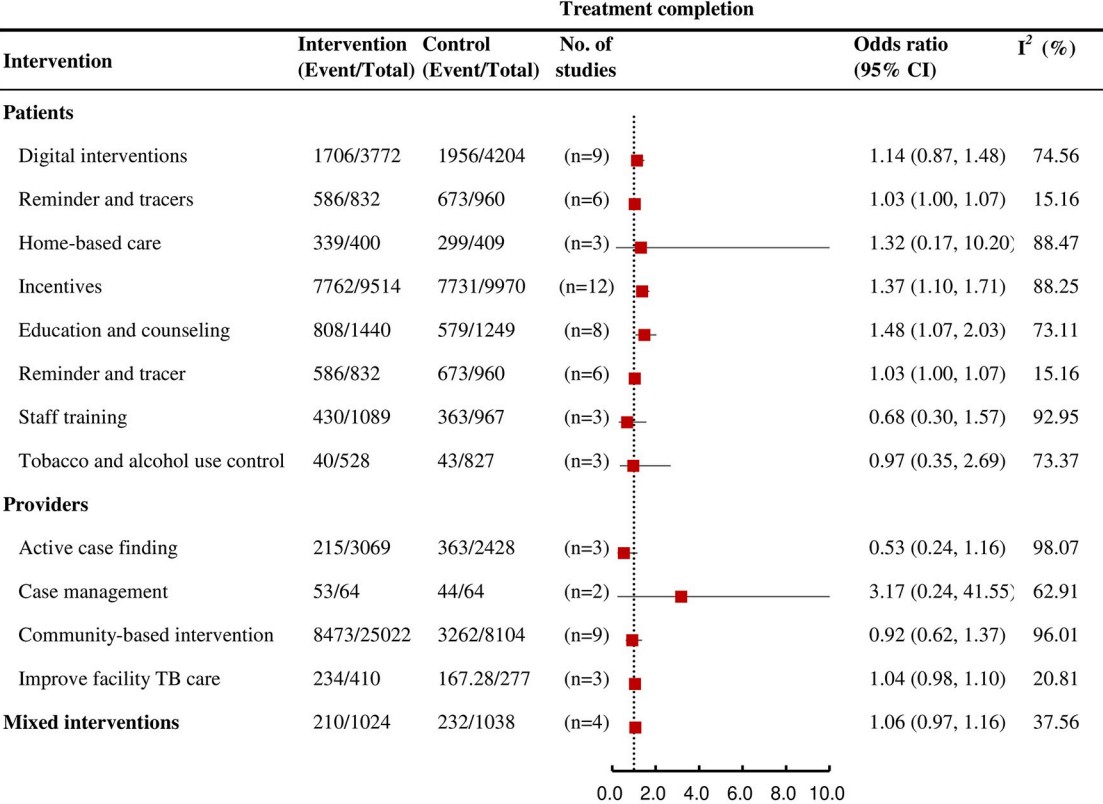

| Intervention | Treatment completion | | | | Odds ratio | $I^2$ (%) |
| | Intervention (Event/Total) | Control (Event/Total) | No. of studies | | (95% CI) | |
| --- | --- | --- | --- | --- | --- | --- |
| **Patients** | | | | | | |
| Digital interventions | 1706/3772 | 1956/4204 | (n=9) | | 1.14 (0.87, 1.48) | 74.56 |
| Reminder and tracers | 586/832 | 673/960 | (n=6) | | 1.03 (1.00, 1.07) | 15.16 |
| Home-based care | 339/400 | 299/409 | (n=3) | | 1.32 (0.17, 10.20) | 88.47 |
| Incentives | 7762/9514 | 7731/9970 | (n=12) | | 1.37 (1.10, 1.71) | 88.25 |
| Education and counseling | 808/1440 | 579/1249 | (n=8) | | 1.48 (1.07, 2.03) | 73.11 |
| Reminder and tracer | 586/832 | 673/960 | (n=6) | | 1.03 (1.00, 1.07) | 15.16 |
| Staff training | 430/1089 | 363/967 | (n=3) | | 0.68 (0.30, 1.57) | 92.95 |
| Tobacco and alcohol use control | 40/528 | 43/827 | (n=3) | | 0.97 (0.35, 2.69) | 73.37 |
| **Providers** | | | | | | |
| Active case finding | 215/3069 | 363/2428 | (n=3) | | 0.53 (0.24, 1.16) | 98.07 |
| Case management | 53/64 | 44/64 | (n=2) | | 3.17 (0.24, 41.55) | 62.91 |
| Community-based intervention | 8473/25022 | 3262/8104 | (n=9) | | 0.92 (0.62, 1.37) | 96.01 |
| Improve facility TB care | 234/410 | 167.28/277 | (n=3) | | 1.04 (0.98, 1.10) | 20.81 |
| **Mixed interventions** | 210/1024 | 232/1038 | (n=4) | | 1.06 (0.97, 1.16) | 37.56 |

0.0  2.0  4.0  6.0  8.0  10.0

**Fig 7. Forest plots showing the effects of various interventions on TB treatment completion outcomes for active TB.**

**TB testing.** According to RCT studies only, digital interventions (OR = 1.97, 95% CI: 1.28 to 3.04; $I^2$ = 83.7%, k = 2) and multisector collaborations (OR = 4.00, 95% CI: 3.37 to 4.75; $I^2$ = 0%, k = 1) were the interventions with effects on multiple care outcomes associated with increased likelihood of testing. Whereas education and counseling (OR = 6.63, 95% CI: 1.11 to 39.55; $I^2$ = 98.4%, k = 4), incentives (OR = 1.74, 95% CI: 1.63 to 1.85; $I^2$ = 45.3%, k = 5) and multisector collaborations (OR = 1.22, 95% CI: 1.08 to 1.38; $I^2$ = 0, k = 1) were interventions with effects on multiple outcomes associated with an increased likelihood of testing among non-RCT studies (S2 Fig).

**TB diagnosis.** Education and counseling (OR = 1.58, 95% CI: 1.09 to 2.29; $I^2$ = 93.5%, k = 7), and active case finding (OR = 1.44, 95% CI: 1.21 to 1.70; $I^2$ = 40.9%, k = 6) and multisector collaborations (OR = 2.33, 95% CI: 1.65 to 3.30; $I^2$ = 0%, k = 1) were associated with increased likelihood of TB diagnosis among RCTs. Comparatively, only multisector collaborations (OR = 10.70, 95% CI: 1.42 to 80.49; $I^2$ = 99.3%, k = 4) was associated with an increased likelihood of diagnosis among non-RCT studies (S2 Fig).

**Linkage-to-care.** Education and counseling (OR = 3.10, 95% CI: 1.97 to 4.86; $I^2$ = 0%, k = 1), digital interventions (OR = 1.10, 95% CI: 1.04 to 1.17; $I^2$ = 26.0%, k = 4), and multisector collaborations (OR = 1.39, 95% CI: 1.04 to 1.86; $I^2$ = 0%, k = 1) were associated with an increased likelihood of linkage-to-care among RCTs only. Among non-RCT studies, incentives (OR = 4.79, 95% CI: 1.28 to 17.94; $I^2$ = 78.4%, k = 2), community-based interventions (OR = 26.93, 95% CI: 16.21 to 44.75; $I^2$ = 19.3%, k = 2), and multisector collaboration (OR = 3.80, 95% CI: 2.73 to 5.28; $I^2$ = 34.49%, k = 2) were each associated with an increased likelihood of linkage-to-care (S2 Fig).

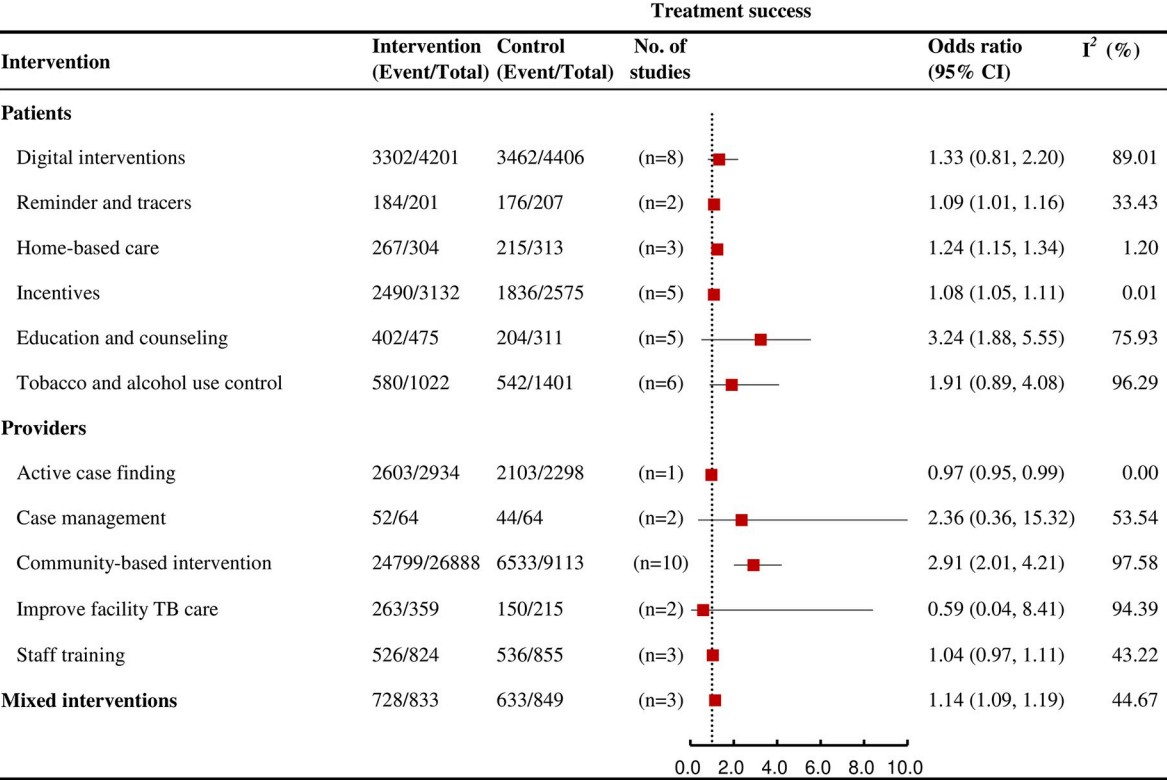

| | | | | Treatment success | |
|---|---|---|---|---|---|
| **Intervention** | **Intervention (Event/Total)** | **Control (Event/Total)** | **No. of studies** | **Odds ratio (95% CI)** | **I² (%)** |
| **Patients** | | | | | |
| Digital interventions | 3302/4201 | 3462/4406 | (n=8) | 1.33 (0.81, 2.20) | 89.01 |
| Reminder and tracers | 184/201 | 176/207 | (n=2) | 1.09 (1.01, 1.16) | 33.43 |
| Home-based care | 267/304 | 215/313 | (n=3) | 1.24 (1.15, 1.34) | 1.20 |
| Incentives | 2490/3132 | 1836/2575 | (n=5) | 1.08 (1.05, 1.11) | 0.01 |
| Education and counseling | 402/475 | 204/311 | (n=5) | 3.24 (1.88, 5.55) | 75.93 |
| Tobacco and alcohol use control | 580/1022 | 542/1401 | (n=6) | 1.91 (0.89, 4.08) | 96.29 |
| **Providers** | | | | | |
| Active case finding | 2603/2934 | 2103/2298 | (n=1) | 0.97 (0.95, 0.99) | 0.00 |
| Case management | 52/64 | 44/64 | (n=2) | 2.36 (0.36, 15.32) | 53.54 |
| Community-based intervention | 24799/26888 | 6533/9113 | (n=10) | 2.91 (2.01, 4.21) | 97.58 |
| Improve facility TB care | 263/359 | 150/215 | (n=2) | 0.59 (0.04, 8.41) | 94.39 |
| Staff training | 526/824 | 536/855 | (n=3) | 1.04 (0.97, 1.11) | 43.22 |
| **Mixed interventions** | 728/833 | 633/849 | (n=3) | 1.14 (1.09, 1.19) | 44.67 |

**Fig 8. Forest plots showing the effects of various interventions on TB treatment success outcomes for active TB.**

**Cure.** Community-based interventions (OR = 2.15, 95% CI: 1.41 to 3.27; $I^2$ = 63.5%, k = 5), mixed interventions (OR = 1.19, 95% CI: 1.13 to 1.26; $I^2$ = 38.8%, k = 3), and education and counseling (OR = 3.45, 95% CI: 1.92 to 6.18; $I^2$ = 0%, k = 2) were associated with increased likelihood of TB cure in RCTs only. Among non-RCT studies, only community-based intervention (OR = 2.92, 95% CI: 2.08 to 4.10; $I^2$ = 95.4, k = 4) and incentives (OR = 1.97, 95% CI: 1.21 to 3.19; $I^2$ = 96.3, k = 3) were each associated with an increased likelihood of TB cure (S2 Fig).

**Treatment completion.** Education and counseling (OR = 1.47, 95% CI: 1.00 to 2.16; $I^2$ = 70.2%, k = 5), and digital health (OR = 1.05, 95% CI: 1.01 to 1.10; $I^2$ = 49.8%, k = 6) in RCT-only studies and incentives (OR = 1.76, 95% CI: 1.19 to 2.62; $I^2$ = 59.5%, k = 5) in non-RCT were each associated with an increased likelihood of treatment completion (S2 Fig).

**Treatment success.** Education and counseling (OR = 4.85, 95% CI: 2.75 to 8.58; $I^2$ = 0%, k = 2), incentives (OR = 1.08, 95% CI: 1.05 to 1.12; $I^2$ = 0%, k = 3), community-based interventions (OR = 2.69, 95% CI: 1.14 to 6.33; $I^2$ = 87.8%, k = 5), and mixed interventions (OR = 1.14, 95% CI: 1.09 to 1.19; $I^2$ = 38.8%, k = 3) were associated with an increased likelihood of treatment success among RCT-only studies. At the same time, education and counseling (OR = 1.28, 95% CI: 1.08 to 1.51; $I^2$ = 0%, k = 3), incentives (OR = 1.83, 95% CI: 1.11 to 3.03; $I^2$ = 45.6%, k = 2), and community-based interventions (OR = 3.38, 95% CI: 2.48 to 4.61; $I^2$ = 93.3%, k = 5) were each associated with an increased likelihood of treatment success in non-RCTs (S2 Fig).

## LMICs vs. HICs

Between LMIC and HIC studies, only five interventions (education and counseling, incentives, community-based intervention, digital interventions, and multisector collaborations) were associated with multiple TB care cascade outcomes (Table 5).

**Table 5. Interventions significantly associated with TB care outcomes at multiple stages of the TB care cascade stratified by subgroups.**

| | TB testing | TB diagnosis | Linkage to care | Treatment completion | Cured | Treatment success |
|---|---|---|---|---|---|---|
| **Intervention (RCT)** | OR [95% CI]; k | OR [95% CI]; k | OR [95% CI]; k | OR [95% CI]; k | OR [95% CI]; k | OR [95% CI]; k |
| Education and counseling | - | 1.58 [1.09–2.29]; k = 7 | 3.10 [1.97–4.86]; k = 1- | 1.47 [1.00–2.16]; k = 5 | 3.45 [1.92–6.18]; k = 2 | 4.85 [2.75–8.58]; k = 2 |
| Incentives | - | - | - | - | - | 1.08 [1.05–1.12]; k = 3 |
| Community-based intervention | - | - | - | - | 2.15 [1.41–3.27]; k = 5 | 2.69 [1.14–6.33]; k = 5 |
| Digital interventions | 1.97 [1.28–3.04]; k = 2 | - | 1.10 [1.04–1.17]; k = 4 | 1.05 [1.01–1.10]; k = 6 | - | - |
| ACF | - | 1.44 [1.21–1.70]; k = 6 | 1.00 [1.00–1.00]; k = 3 | - | - | - |
| Mixed interventions | - | - | - | - | 1.19 [1.13–1.26]; k = 3 | 1.14 [1.09–1.19]; k = 3 |
| Multisector collaborations | 4.00 [3.37–4.75]; k = 1 | 2.33 [1.65–3.30]; k = 1 | 1.39 (1.04–1.86); k = 1 | - | - | - |
| **Intervention (non-RCTs)** | | | | | | |
| Education and counseling | 6.63[1.11–39.55]; k = 4 | - | - | - | - | 1.28 [1.08–1.51]; k = 3 |
| Incentives | 1.74[1.63–1.85]; k = 5 | - | 4.79 [1.28–17.94]; k = 2 | 1.76 [1.19–2.62]; k = 5 | 1.97 [1.21–3.19]; k = 3 | 1.83 [1.11–3.03]; k = 2 |
| Community-based intervention | - | - | 26.93[16.21–44.75]; k = 2 | - | 2.92 [2.08–4.10]; k = 4 | 3.38 [2.48–4.61]; k = 5 |
| Multisector collaborations | 1.22[1.08–1.38]; k = 1 | 10.70[1.42–80.49]; k = 4 | 3.80 [2.73–5.28]; k = 2 | - | - | - |
| **Intervention (LMICs)** | | | | | | |
| Education and counseling | 16.48[4.40–61.70]; k = 3 | 1.44 [1.08–1.92]; k = 9 | 3.10[1.97–4.86]; k = 1 | 1.17 [1.04, 1.32]; k = 5 | 2.08 [1.11–3.88]; k = 4 | - |
| Incentives | - | - | 2.27[1.55–3.33]; k = 2 | 1.44 [1.04, 2.00]; k = 9 | 1.62 [1.06–2.48]; k = 5 | 1.08 [1.05–1.11]; k = 5 |
| Community-based intervention | - | - | 9.91[1.86–52.74]; k = 4 | - | 2.53 [1.92–3.35]; k = 9 | 2.91[2.01–4.21]; k = 10 |
| Digital interventions | 2.17[1.67–2.82]; k = 1 | | 1.10 [1.04–1.17]; k = 4 | - | - | - |
| Multisector collaborations | 4.00[3.37–4.75]; k = 1 | 8.00[1.53–41.84]; k = 5 | 3.25[2.05–5.14]; k = 3 | - | - | - |
| Mixed interventions | | | | | 1.19 [1.13–1.26]; k = 3 | 1.14 [1.09–1.19]; k = 3 |
| **Intervention (HICs)** | | | | | | |
| Incentives | 1.74[1.63–1.85]; k = 5 | - | - | 1.04 [1.02–1.07]; k = 3 | - | - |
| **Intervention (HIV-integrated)** | | | | | | |
| Education and counseling | 10.06[1.69–59.78]; k = 3 | 1.25 [1.09–1.44]; k = 2 | 3.10[1.97–4.86]; k = 1 | - | - | - |
| **Intervention (not HIV-integrated)** | | | | | | |
| Education and counseling | | 1.56[1.03–2.35]; k = 7 | | 1.48 [1.07–2.03]; k = 8 | 2.08 [1.11–3.88]; k = 4 | 3.24[1.88–5.55]; k = 5 |
| Incentives | 1.78[1.66–1.90]; k = 4 | - | 2.86[1.25–6.50]; k = 4 | 1.37 [1.10–1.71]; k = 12 | 1.62 [1.06–2.48]; k = 5 | 1.08 [1.05–1.11]; k = 5 |
| Community-based intervention | - | - | - | - | 2.53 [1.92–3.35]; k = 9 | 2.91[2.01–4.21]; k = 10 |

(*Continued*)

**Table 5.** (Continued)

| | TB testing | TB diagnosis | Linkage to care | Treatment completion | Cured | Treatment success |
|---|---|---|---|---|---|---|
| Digital interventions | 1.58[1.35–1.87]; k = 1 | - | 1.12[1.04–1.20]; k = 2 | - | - | - |
| Home-based care | - | - | - | 1.17 [1.11–1.24]; k = 2 | - | 1.24[1.15–1.33]; k = 2 |
| Mixed interventions | - | - | - | - | 1.19 [1.13–1.26]; k = 3 | 1.14[1.09–1.19]; k = 3 |
| Multisector collaborations | 4.00[3.37–4.75]; k = 1 | 8.00 [1.53–41.84]; k = 4 | 3.25[2.05–5.14]; k = 3 | - | - | - |

**ACF,** active case finding; **OR**, odds ratio; **CI,** confidence interval; **k,** number of studies.

-: No study assessed how this intervention affected this outcome or the effects of this intervention on this outcome was not statistically significant.

## Testing

In LMIC studies, education and counseling (OR = 16.48, 95% CI: 4.40 to 61.70; $I^2$ = 97.4%, k = 3), digital interventions (OR = 2.17, 95% CI: 1.67 to 2.82; $I^2$ = 0%, k = 1), and multisector collaborations (OR = 4.00, 95% CI: 3.37 to 4.75; $I^2$ = 0%, k = 1) were associated with an increased likelihood of TB testing. At the same time, only incentives (OR = 1.74, 95% CI: 1.63 to 1.85; $I^2$ = 45.3%, k = 5) as an intervention with multistage effect was associated with an increased likelihood of TB testing in HIC studies (S3 Fig).

**Diagnosis.** Only multisector collaboration (OR = 8.00, 95% CI: 1.53 to 41.84; $I^2$ = 99.3%, k = 5) and education and counseling (OR = 1.44, 95% CI: 1.08 to 1.92; $I^2$ = 89.9%, k = 9) were associated with increased likelihood of diagnosis in LMIC-only studies. But none of the interventions with effect on multiple care cascade outcomes was associated with diagnosis in HIC studies (S3 Fig).

**Linkage-to-care.** Similar to TB testing, education and counseling (OR = 3.10, 95% CI: 1.97 to 4.86; $I^2$ = 0%, k = 1), incentives (OR = 2.27, 95% CI: 1.55 to 3.33; $I^2$ = 0%, k = 2), community-based interventions (OR = 9.91, 95% CI: 1.86 to 52.74; $I^2$ = 93.8%, k = 4), digital interventions (OR = 1.10, 95% CI: 1.04 to 1.17; $I^2$ = 26.0%, k = 4), and multisector collaborations (OR = 3.25, 95% CI: 2.05 to 5.14; $I^2$ = 68.9%, k = 3) were associated with increased likelihood of linkage-to-care in LMIC studies. But none of the interventions with effect on multiple care cascade outcomes was associated with linkage to care in HIC studies (S3 Fig).

**Cure.** In LMICs, education and counseling (OR = 2.08, 95% CI: 1.11 to 3.881.86–3.58; k = 4), incentives (OR = 1.62, 95% CI: 1.06 to 2.48; $I^2$ = 97.2%, k = 5), mixed interventions (OR = 1.19, 95% CI: 1.13 to 1.26; $I^2$ = 38.8%, k = 3), and community-based interventions (OR = 2.53, 95% CI: 1.92 to 3.35; $I^2$ = 91.8%, k = 9) were associated with increased likelihood of TB cure. However, not interventions were found to be associated with cure in HIC studies (S3 Fig).

**Treatment completion.** Education and counseling (OR = 1.17; k = 5, 95% CI: 1.04, 1.32; $I^2$ = 0%, k = 5) and incentives (OR = 1.531.44, 95% CI: 1.04, 2.00; $I^2$ = 77.8, k = 9) were each associated with an increased likelihood of treatment completion in LMIC studies. While only incentives (OR = 1.04, 95% CI: 1.02 to 1.07; $I^2$ = 18.6, k = 3) was associated with an increased likelihood of treatment completion in HIC studies (S3 Fig).

**Treatment success.** In LMIC studies, community-based interventions (OR = 2.91, 95% CI: 2.01 to 4.21; $I^2$ = 94.7, k = 10) and mixed interventions (OR = 1.14, 95% CI: 1.09 to 1.19; $I^2$ = 38.8, k = 3) were associated with an increased likelihood of treatment success. However, none of the interventions with effects on multiple TB care cascade outcomes were associated with an increased likelihood of treatment success in HIC studies (S3 Fig).

### HIV integration vs. non-HIV integration

Among studies appraising TB interventions integrated with HIV services and non-HIV integrated interventions, education and counseling, incentives, and multisector collaborations were associated with at least three TB care cascade outcomes (Table 5).

**Testing.** Among HIV-integrated TB interventions, education and counseling (OR = 10.06, 95% CI: 1.69 to 59.78; $I^2$ = 98.9, k = 3) was the only intervention with multistage effect associated with and increased likelihood of testing. But among non-HIV integrated studies, digital health (OR = 1.58, 95% CI: 1.35 to 1.58; $I^2$ = 0%, k = 1), multisector collaborations (OR = 4.00, 95% CI: 3.37 to 4.75; $I^2$ = 0%, k = 4), and incentives (OR = 1.78, 95% CI: 1.66 to 1.90; k = 4) were associated with an increased likelihood of testing (S4 Fig).

**Diagnosis.** Only education and counseling (OR = 1.25, 95% CI: 1.09 to 1.44; $I^2$ = 0%, k = 2) was associated with an increased likelihood of TB diagnosis among HIV-integrated interventions. At the same time, multisector collaboration (OR = 8.00, 95% CI: 1.53 to 41.84; $I^2$ = 99.3%, k = 4) and education and counseling (OR = 1.56, 95% CI: 1.03 to 2.35; $I^2$ = 89.3, k = 7) were the non-HIV-integrated interventions associated with an increased likelihood of TB diagnosis (S4 Fig).

**Linkage-to-care.** Similarly, only education and counseling (OR = 3.10, 95% CI: 1.97 to 4.86; $I^2$ = 0%, k = 1) as a multistep effective intervention was associated with an increased likelihood of linkage to care among HIV-integrated interventions, while multisector collaboration (OR = 3.25, 95% CI: 2.05 to 5.14; $I^2$ = 68.9, k = 3), digital health (OR = 1.12, 95% CI: 1.04 to 1.20; $I^2$ = 0%, k = 2), and incentives (OR = 2.86, 95% CI: 1.25 to 6.50; $I^2$ = 77.7%, k = 4) were associated with an increased likelihood of linkage-to-care among nonintegrated interventions (S4 Fig).

**Cured.** Although none of the interventions with multistage effects were associated with cure outcomes among HIV-integrated studies, community-based interventions (OR = 2.53, 95% CI: 1.92 to 3.35; $I^2$ = 91.8, k = 9), mixed interventions (OR = 1.19, 95% CI: 1.13 to 1.26; $I^2$ = 38.8%, k = 3), counseling and education (OR = 2.08, 95% CI: 1.11 to 3.88; $I^2$ = 41.8%, k = 4), and incentives (OR = 1.62, 95% CI: 1.06 to 2.48; $I^2$ = 97.2%, k = 5) were associated with increased likelihood of TB cure among nonintegrated TB interventions (S4 Fig).

**Treatment completion.** Like cure, no interventions with multistage effects were associated with treatment completion among HIV-integrated studies. However, counseling and education (OR = 1.48, 95% CI: 1.07 to 2.03; $I^2$ = 57.5%, k = 8), home-based care (OR = 1.17, 95% CI: 1.11 to 1.24; $I^2$ = 10%, k = 2), and incentives (OR = 1.37, 95% CI: 1.10 to 1.71; $I^2$ = 68.5%, k = 12) were associated with increased likelihood of treatment completion in nonintegrated studies (S4 Fig).

**Treatment success.** Similarly, no interventions with multistage effects were associated with treatment success among HIV-integrated studies. At the same time, home-based care (OR = 1.24, 95% CI: 1.15 to 1.33; $I^2$ = 37.3%, k = 2), incentives (OR = 1.08, 95% CI: 1.05 to 1.11; $I^2$ = 31.5%, k = 5), education and counseling (OR = 3.24, 95% CI: 1.88 to 5.55; $I^2$ = 37.8%, k = 5), community-based interventions (OR = 2.91, 95% CI: 2.01 to 4.21; $I^2$ = 94.7%, k = 10), and mixed interventions (OR = 1.14, 95% CI: 1.09 to 1.19; $I^2$ = 38.8%, k = 3) were associated with an increased likelihood of treatment success in nonintegrated studies (S4 Fig).

## Discussion

Ensuring the delivery of quality person-centered service to all people living with TB is a global TB control priority and crucial to ending the TB pandemic [117,118]. This review synthesized existing evidence on the effects of various TB interventions in optimizing care cascade outcomes from a global perspective. Our findings extend the literature by summarizing evidence on how the intervention impacts TB care cascade outcomes to inform holistic TB control

strategies. Among TB interventions, education and counseling were associated with an increased likelihood of TB testing, diagnosis, cure, treatment completion, and treatment success compared to standard-of-care. Mixed interventions, community-based interventions, and incentives were each associated with multiple care cascade outcomes, and digital interventions were significantly associated with two care cascade outcomes.

Per our findings, community-based interventions, incentives, and multisector collaborations were the interventions associated an increased likelihood of outcomes for at least three care cased stages in LMICs. The evidence quality ranged from low to moderate certainty GRADE assessment. This is consistent with results in the literature on community-based interventions associated with testing, linkage-to-care, and treatment adherence [119–123]. Our sub-analysis findings showed that community-based interventions increased the likelihood of testing, especially in LMICs, where TB is estimated to be more prevalent. Community-based interventions and multisector collaborations may have worked well in LMICs due to the fragile and fragmented healthcare systems and limited resources of many LMICs [124,125]. More research is needed on how best to implement and fund community-based TB care to improve overall outcomes in the settings.

The overall and RCTs-only sub-analysis showed that education and counseling increased the likelihood of all TB care cascade outcomes with an average moderate certainty. Mitigating public misconceptions and stigma that hinder TB services utilization through education and counseling may explain this observation. Similar to our findings, a previous review found that patient counseling at diagnosis improved linkage-to-care and treatment completion among TB patients [126]. Another review and Ethiopian study found that education and counseling engaged TB patients in the active self-management of their TB infection, which was essential to cure, treatment success, and the reduction of self-stigma [127,128]. We, therefore, emphasize that education and counseling should be valued and incorporated as a necessary component of ending TB strategies, especially for newly diagnosed TB patients.

Mixed interventions and incentives were associated with an increased likelihood of multiple care cascade outcomes with low to moderate evidence quality. Some studies combined two or more interventions and reported the effect of this tailored intervention on TB care. However, it is worth noting that various factors (including where binding constraints are on the cascade, the overall effectiveness of interventions mixed at each point, and the relative costs of different interventions) influence the impact of mixed interventions. Therefore, researchers should consider mixing interventions with multiple outcome effects or efficient single-step strategies that span all six steps of the care cascade when designing mixed interventions. Also, resource availability, existing structures, local settings, and long-term sustainability should be well thought-out when adopting methods like incentives in limited resources settings.

Overall, digital intervention only increased the likelihood of testing and linkage to care. This finding conforms with a previous review's results that evidence of digital technologies improving TB care is contradictory and limited [129]. However, digital interventions have significantly improved healthcare services delivery and uptake for other infectious diseases like HIV [130]. New digital tools like smartphone-based diagnostics are cost-effective for rapid diagnostics in point-of-care testing and could enable real-time remote patient monitoring [131,132]. Thus, the roles of digital interventions in decentralizing and expanding healthcare could be tailored for efficient use in TB care and should be further researched.

To be noted, the interventions identified in this review were highly diverse. The pooled effects of these interventions on TB care cascade were highly heterogenous. Results from our meta-regression analysis showed that the study design, year of publication, and region of study were the major sources of heterogeneity. This was not surprising as we observed wide

diversification in how interventions were designed, the duration and intensity of implementation and variations in how the interventions were implemented. The variations in settings (rural communities versus urban slums) and target populations (general populations versus prisoners or ex-convicts) and approach to implementation for each intervention contributed to heterogeneity. For example, a study adapted "staff training + recruited and trained lay workers + active case finding" as a comprehensive care approach, while another study implemented "peer training + patient counseling and education +onsite sputum collection + expedited treatment initiation" to improve case detection and treatment outcomes [133,134]. Therefore, attention should be paid to local setting needs and cultural context when choosing to adopt any of the interventions with multistage effects to improve TB care cascade outcomes.

We observed some evidence of publication bias per the funnel plots among the studies assessing interventions in TB diagnosis and treatment completion. This suggests that the effect size of certain interventions found in under this outcome may have been affected by missing small-size or negative finding studies. The reluctance of academic journals to publish studies with negative findings and our exclusion of case reports and short research reports may have contributed to this bias. Therefore, our reported effects of interventions on multiple care cascade outcomes should be interpreted with caution and within context. However, we corrected this bias by adjusting the plot using the trim-and-fill method. Therefore, our drawn conclusions based on the meta-analysis results remain salient.

Our study has implementation, policy, and research implications. First, our findings reiterate the WHO recommendation that education and counseling should form part of comprehensive TB care strategies [135]. Thus, countries without patient counseling guidelines should consider establishing policies to incorporate counseling into routine TB care. Secondly, merging or concurrently implementing intervention with multiple outcomes effect could improve global TB control significantly. Therefore, researchers should consider revising mixed TB interventions to incorporate more such interventions or effective single-step strategies that span the entire care cascade. Also, programs to upscale evidence-based approaches should consider local context variations and adjust strategies to reach national TB goals. Finally, digital health is the cornerstone of modern healthcare but have unclear impact on TB care cascade outcomes. Therefore, future research should further explore potential roles of digital intervention in optimizing TB care.

Our study has some limitations. First, the interventions were highly heterogeneous due to many factors, and the differences in implementation approaches like intensity, coverage, local context settings, and resource availability may have contributed to their effectiveness. Second, we did not evaluate the cost of implementing these interventions. Hence, our findings should be interpreted with attention to cost and feasibility. We also excluded non-English studies, which may have impacted the capture of literature from bibliographic databases of non-anglophone countries and biased our findings. However, our findings can be generalized because many of the interventions reviewed targeted diverse populations like PLWH, children, and migrants in both HIC and LMIC settings. Third, our study does not cover the entire TB care cascade as gaps in the early stages (focusing on testing or pretreatment loss to follow-up) and post-treatment outcomes (like TB recurrence and death) were not outcomes of interest and were not assessed in this review. Future reviews should consider examining these gaps and other key distinctions (like drug susceptibility or different forms of TB) and their effects on TB care outcomes to help inform strategies and policy adoption. Fourth, our funnel plots and meta-regression results showed the existence of publication bias. Nonetheless, our findings remain relevant to informing TB intervention programming if interpreted with caution and within context.

## Conclusions

Our study shows the existence of a wide range of relatively simple interventions that could substantially improve TB care outcomes. Nonetheless, high fidelity along the care cascade would become increasingly important as the rate of TB drug resistance increases. Therefore, multi-step efficient interventions like education and counseling, incentives, and mixed interventions should be keenly considered in expanding active TB control programs. But factors like differences in implementation intensity, resource availability, and local setting contexts should be well thought-out when choosing strategies to strengthen holistic TB care as the interventions were sufficiently heterogeneous.

## Supporting information

**S1 File. Supporting information file. Protocol A in S1 File.** Study protocol as published on PROSPERO. **Table B in S1 File.** Search strategy and results. **Table C in S1 File.** Characteristics of included studies grouped by intervention type. **Table D in S1 File.** Results of meta-regression analysis. **Table E in S1 File.** Outcome of risk of bias assessment for 84 RCTs studies using the Cochrane Risk of Bias Assessment Tool. **Table F in S1 File.** Quality assessment of included studies using the EPHPP quality assessment tool. **Table G in S1 File.** GRADE Outcomes. **Table H in S1 File**. PRISMA checklist for protocol.
(DOCX)

**S1 Fig. Funnel plots.**
(TIFF)

**S2 Fig. Forest plots of subgroup analysis by study design (RCTs vs. non-RCTs).**
(TIFF)

**S3 Fig. Forest plots of subgroup analysis by region (LIMCs vs. HICs).**
(TIFF)

**S4 Fig. Forest plots of subgroup analysis by services integration (LIMCs vs. HICs).**
(TIFF)

**S1 Data. Data sets used in this review stratified by cascade outcomes.**
(XLSX)

**S2 Data. Spreadsheet used in data extraction with details of all data extracted for each study.**
(XLSX)

## Acknowledgments

We appreciate Dr. Madhu Pai, Dr. Liu Wei, and Miss. Jiayu He for serving as scientific advisors during this study. We also thank Miss Shamen Susan Chauma for assisting with screening articles and extracting data.

## Author Contributions

**Conceptualization:** Joseph D. Tucker, Weiming Tang.

**Data curation:** Gifty Marley, Juan Nie, Weibin Cheng, Huipeng Liao.

**Formal analysis:** Xia Zou, Juan Nie, Weibin Cheng.

**Investigation:** Weiming Tang.

**Methodology:** Gifty Marley, Xia Zou, Juan Nie, Weibin Cheng, Yewei Xie, Huipeng Liao, Yehua Wang.

**Resources:** Weiming Tang.

**Supervision:** Joseph D. Tucker, Sean Sylvia, Roger Chou, Dan Wu, Jason Ong, Weiming Tang.

**Validation:** Yewei Xie, Huipeng Liao, Yehua Wang, Yusha Tao, Joseph D. Tucker, Sean Sylvia, Roger Chou, Dan Wu, Jason Ong, Weiming Tang.

**Writing – original draft:** Gifty Marley, Weiming Tang.

**Writing – review & editing:** Gifty Marley, Xia Zou, Juan Nie, Weibin Cheng, Yewei Xie, Huipeng Liao, Yehua Wang, Yusha Tao, Joseph D. Tucker, Sean Sylvia, Roger Chou, Dan Wu, Jason Ong, Weiming Tang.

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
