## [Editor Report · Decision Letter 0]

15 Aug 2022

Dear Dr Tang, 

Thank you for submitting your manuscript entitled "Improving Continuum Outcomes for Active TB: A Global Systematic Review and Meta-Analysis of TB Interventions." for consideration by PLOS Medicine.

Your manuscript has now been evaluated by the PLOS Medicine editorial staff and I am writing to let you know that we would like to send your submission out for external peer review.

Please re-submit your manuscript within two working days, i.e. by Aug 17 2022 11:59PM.

Kind regards,

Caitlin Moyer, Ph.D.

Associate Editor

PLOS Medicine

---

## [Decision Letter · Decision Letter 1]

28 Sep 2022

Dear Dr. Tang,

Thank you very much for submitting your manuscript "Improving Continuum Outcomes for Active TB: A Global Systematic Review and Meta-Analysis of TB Interventions." (PMEDICINE-D-22-02722R1) for consideration at PLOS Medicine. 

Your paper was evaluated by an associate editor and discussed among all the editors here. It was also discussed with an academic editor with relevant expertise, and sent to independent reviewers, including a statistical reviewer. The reviews are appended at the bottom of this email and any accompanying reviewer attachments can be seen via the link below:

[LINK]

In light of these reviews, I am afraid that we will not be able to accept the manuscript for publication in the journal in its current form, but we would like to consider a revised version that addresses the reviewers' and editors' comments. Obviously we cannot make any decision about publication until we have seen the revised manuscript and your response, and we plan to seek re-review by one or more of the reviewers. 

We hope to receive your revised manuscript by Oct 19 2022 11:59PM. Please email us (plosmedicine@plos.org) if you have any questions or concerns.

We look forward to receiving your revised manuscript. 

Sincerely,

Callam Davidson

PLOS Medicine

plosmedicine.org

Abstract:

* Please ensure that all numbers presented in the abstract are present and identical to numbers presented in the main manuscript text.

* Please combine the Methods and Findings sections into one section, “Methods and findings”.

* Please include the study design, dates of search, types of study designs included, and eligibility criteria. 

The Financial Disclosure should describe sources of funding that have supported the work. If your manuscript is published, your statement will appear in the Funding section of the article. See https://journals.plos.org/plosmedicine/s/submission-guidelines#loc-financial-disclosure-statement for more information. Please remove the ‘Funding’ section from the main text and instead include this information in the Financial Disclosure section of the Submission Form. 

Please remove the ‘Summary’ section and instead include a short, non-technical Author Summary of your research to make findings accessible to a wide audience that includes both scientists and non-scientists. The Author Summary should immediately follow the Abstract in your revised manuscript. This text is subject to editorial change and should be distinct from the scientific abstract. Please see our author guidelines for more information: https://journals.plos.org/plosmedicine/s/revising-your-manuscript#loc-author-summary

Please include continuous line numbering throughout your manuscript. 

Please place citations in square rather than round brackets.

Please label items in the Supporting Information as described here: https://journals.plos.org/plosmedicine/s/supporting-information

Please update your search to the present time.

Please provide the beginning and end dates of your search in the Methods.

Please include non-English language sources of studies.

Please remove the Contributors, Conflicts of interest, and Data sharing sections from your main text – all of this information ought to be included in your responses to the relevant parts of the submission form. 

Thank you for reporting your SR/MA according to the PRISMA guidelines. Please update the checklist to use section and paragraph numbers, rather than page numbers.

You note in your ‘in text’ Data sharing section that further data are available on request from the corresponding author. PLOS does not permit the corresponding author to be the main point of contact for data requests. Please relocate the data sharing statement to the relevant part of the submission form and see https://journals.plos.org/plosmedicine/s/data-availability for further information on how best to make your data available. 

Comments from the reviewers:

Reviewer #1: See attachment

Michael Dewey

Reviewer #2: PMEDICINE-D-22-02722R1: Improving Continuum Outcomes for Active TB: A Global Systematic Review and Meta-Analysis of TB Interventions. 

Overall, the systematic review and metanalysis on this topic is highly relevant and I must congratulate the authors for taking this up. 

The author should have given continuous line numbers in their manuscript file so that would have helped the reviewer in indicating the line numbers while making the comments. Nevertheless, the following are the major comments from me. 

1. The major problem with this manuscript is that several aspects of methodology and the results are available in the supplementary file and not as main manuscript. This needs to be corrected.

2. The literature review was last updated till April 2021. The authors are requested to update their systematic review and include any new studies that have been published, so that their manuscript is up to date. 

3. The specific objectives highlighting the PICO (target patients, interventions, control population, outcomes) elements for each step/multiple steps in the care continuum needs to be spelt out in the main manuscript. These are currently seen only in the supplementary tables.

4. The search terms must be derived from these objectives for each PICO element. The search terms mentioned under methods is not of sufficient detail and the exact terms and search methodology for searching for articles in each of the data bases needs to be mentioned very clearly. These are listed in the supplementary files and needs to be brought out in the main manuscript. 

5. The operational definition given in Table 1 of what is considered as an intervention and what is not considered as an intervention in this systematic review needs to be spelt out. For example, if the intervention includes a diagnostic tool such as Xpert TB/Rif® test introduction or introduction of a newer TB treatment regimen, whether this is considered an intervention to improve TB care continuum is unclear. 

6. The description of the intervention that was picked up from each study needs to be more elaborate. For example, describing an intervention as "Education & Counselling" without indicating what are the element of this intervention is not useful to understand this intervention and replicate it in future. Also, regarding interventions, was there a dose response relationship? More intensive education and counselling versus less intensive education and counselling? This sort is description is needed for each intervention described in this manuscript. 

7. The quality of all the figures/images is very poor and the text is not easily readable. This needs to be replaced. 

Reviewer #3: This is a very large systematic review that makes two innovative contributions to the global TB literature. First, most systematic reviews of TB care interventions focus on TB treatment, whereas numerous studies of the care cascade have shown that the largest losses of people with TB occur well before individuals get to initiate TB treatment. As such, by conducting a review that evaluates interventions across multiple stages of the care cascade, the authors go beyond the scope of most systematic reviews in this space and address stages that receive less attention. Second, evaluating interventions for all of these stages together allows for comparison and provides important insights. For example, the most notable finding—regarding the potential benefits of counseling across multiple stages of care, is very helpful to TB programs, because it allows for focused investment in this intervention to address multiple gaps in care.

With that said, the challenge with this review is dealing with the problem of heterogeneity—by study setting (e.g., high-income vs. lower income countries), patient population, and with regard to the interventions themselves, among other forms of heterogeneity. The authors have done a reasonable job of dealing with the former two problems but do not provide enough information for readers to understand what these interventions across the various studies might involve. This is a shortcoming of the paper and limits the insights if provides—but I feel that the authors could address this shortcoming by providing a table or tables that provide more specific insights into the types of interventions that were found to be potentially impactful a cross various gaps.

Another major concern is regarding the authors' search strategy—which does not seems sufficiently broad to cover all cascade gaps, as discussed further below.

Major feedback:

1. Search strategy does not seem sufficiently broad: A major concern is that the authors' search strategy does not appear sufficiently broad to actually cover care cascade gaps outside of the TB treatment phase. In fact, I worry that it is not adequate to even cover all interventions in the treatment phase. The umbrella search terms all seem focused on medication adherence—e.g., "adherence", "compliance", "retention", etc. None of these seems appropriate to cover early stages in the cascade focused on testing or pretreatment loss to follow-up. For example, there is no term for "testing", "diagnosis", "linkage to care", "loss to follow-up" (and related variants, such as "initial default", the old term for pretreatment loss to follow-up) which would address gaps in testing or pretreatment loss to follow-up, or on-treatment loss to follow-up for that matter. I am in fact worried that the authors may have initially aimed to conduct a review of interventions for medication adherence and then repurposed the search findings to look across the care cascade. The authors need to present evidence to suggest that gaps in testing, diagnosis, and linkage to care were actually somehow covered in their search, otherwise I'm worried this is a major shortcoming of this review as it's unclear how they would have adequately covered this literature.

2. Definition of "testing" does not cover important testing modalities and confusion between "diagnosis" and "testing": The authors' definition of "testing" (page 7, first lines under "Definitions") does not appear to address Xpert MTB/RIF (or other cartridge-based nucleic acid amplification testing methods such as TrueNat) or mycobacterial culture. It also does not address testing for extrapulmonary TB. It is unclear why some of these modalities fall under the "diagnosis" gap rather than the "testing" gap. For example, how is the process of "administering sputum smear", which counts as testing, any different from the process of administering Xpert MTB/RIF for upfront testing? This raises the question of whether the "testing" and "diagnosis" gaps are really a single gap - as is articulated in most TB care cascades. Again, a contributing factor to this confusion is that it is unclear what the interventions included in the review actually entail—and therefore how addressing "testing" vs. "diagnosis" is different.

3. Need to provide more insights into what interventions actually entail. While the authors do define the broad intervention categories they use, I constantly found myself wondering whether more specific information could be provided on specific characteristics of the interventions to make this review more useful. For example, what were characteristics of the "community-based interventions" or "mixed interventions" that positive impacted outcomes for each outcome? Just as the authors provide a Forest Plot for each outcome, it would greatly increase the value of this review to supply one table per outcome in the main manuscript that provides more details on the specific characteristics of interventions based on the original paper. For example, for "testing" there were 2 studies involving digital interventions, 7 studies involving incentives, 9 studies involving counseling, 6 studies involving active case finding, and 2 studies involving multi-sector collaboration. A single table could very briefly summarize the specific interventions used in each study (with one row per intervention type, e.g., "incentives"). This would allow readers to gain a more specific understanding of what these interventions actually are.

+

4. Need to discuss implications of the funnel plot findings: The authors note at the end that they could not address publication bias - but they actually partly did through the funnel plots, which do suggest evidence of bias. The authors should, in their limitations section, instead address the insights from the funnel plots. Skewed funnel plots usually suggest that studies that had small sample sizes with negative findings may not have been published—which might suggest that the effect sizes seen for certain intervention types might be larger than they would be if such studies had been published and were included. If their funnel plots do suggest that small, negative studies might be missing, they should discuss this point for each outcome and in the Discussion section of the paper.

Minor feedback:

1. Review does not address some major gaps in the care cascade: The authors need to be clear that their review does not actually cover the entire care cascade. For example, although the authors start from "testing" as the first stage, this assumes that patients have made it to clinic sites where testing might be offered. However, across most published country care cascades (e.g., India, South Africa, Zambia, Madagascar, for example), a substantial drop off before testing is partly or mostly attributable to people with TB in the population not even making it to health centers for testing. As such, the authors should be clear that their review does not address this major early problem in the care cascade.

Second, the authors do not address post-treatment TB recurrence and death as a problem. Guidelines, including ones cited by the author, and some country care cascades (e.g., India, Madagascar) point out that a meaningful proportion of patient experience post-treatment TB recurrence or death, especially in the year following treatment completion. While much of this gap is attributable to the quality of TB treatment itself (e.g., whether there is high medication adherence), it is very possible that interventions in the immediate time period after TB treatment could help reduce post-treatment death, for example. As such, post-TB care is a major area of interest for the TB community right now. The authors should also note that their review does not address this gap at all.

Reviewer #4: 

RE: PLOS Medicine 22-02722R1 - comments to the authors 

The authors have done a very extensive review, with very careful and meticulous analysis including careful assessment of quality, , sources of heterogeneity, publication bias, and notably have applied the full-grade criteria to their findings in order to fully assess the quality of evidence for each intervention and each outcome.

I have a number of suggestions for the authors to consider, that I believe would potentially improve the paper and also make the findings a little more accessible, as right now, there is a great deal of information, but I found it difficult to find the most important messages.

Introduction: I wonder if they would consider calling this the TB care cascade - a term that is increasingly used. Also in this section, and later in definitions, they define "linkage to care" as the link between diagnosis and treatment. this is an important linkage but really linkage to care usually refers to the first step - meaning getting individuals to have initial testing. That requires very different approaches compared to linking services within health systems of diagnosis and treatment. 

On page 6 "pharmaceutical studies were excluded…." - I could not understand this sentence.

Quality - they have done an excellent job of rating quality, but it is less clear how they have used the quality ratings, (eg in the analysis); this should be stated in the methods.

Pooling - Top of page 8 "only data...", this sentence is unclear to me and need some clarification. 

Results - In general, the results are dense; I found it difficult reading through it all to get a clear picture of the most important interventions, particularly ones that will act at a number of different steps in the TB care cascade. I think a summary table, similar in structure/concept to supplemental table 5, would be helpful for the main results. I would like to see a table that summarizes the main interventions and their effect estimates at the main steps in the TB care cascade. An alternative is to have figures which show single interventions and how they act at many different cascade steps. From a programmatic point of view, I think most programs are interested not in correcting a single problem, but rather in the entire TB care continuum, so interventions that act across all steps in the continuum are potentially more valuable than those that act only at a particular step. Hence, some conceptualization, potentially a figure or two showing how some of the major interventions, particularly those that were evaluated at various steps, have produced effects. For example, an intervention that results in more individuals getting tested but has no impact in the number of individuals completing treatment successfully is not necessarily that useful, and in fact is likely to be quite cost inefficient since all the costs are added for diagnostic activities and evaluation of patients result in no real benefit, either to individuals or programs. On the other hand, an intervention that has more modest effects but effects are seen throughout the care cascade, so that more patients are not only diagnosed but actually successfully complete treatment is likely to be much more beneficial intervention. But I cannot really get this from the information as it is now presented. 

Along these lines, were there any studies that looked at multiple steps and could these be another sub analysis, i.e., studies that looked at all steps for a given intervention or studies that looked at multiple steps.

A minor point is that I think they should round the estimates and confidence intervals to a maximum of one decimal place. To see an estimate of 16.48 with a confidence interval that ranges from near 1 .0 to over 50 is, in my mind, "pseudo precision". It is a minor point but going to one decimal point or even full numbers would be more accurate and would make the tables and figures a bit more readable.

Finally, but important:

The question about data availability; I do not believe the authors have filled this section correctly. Basically, they say that all data is available in the annex, but this cannot be true. I suggest that their metadata, i.e., really everything abstracted, is placed somewhere that is web-accessible, and the link given in the final version of the manuscript.

[LINK]

---

## [Decision Letter · Decision Letter 2]

30 Nov 2022

Dear Dr. Tang,

Thank you very much for re-submitting your manuscript "Improving Cascade Outcomes for Active TB: A Global Systematic Review and Meta-Analysis of TB Interventions" (PMEDICINE-D-22-02722R2) for review by PLOS Medicine.

I have discussed the paper with my colleagues and the academic editor and it was also seen again by two reviewers. I am pleased to say that provided the remaining editorial and production issues are dealt with we are planning to accept the paper for publication in the journal.

[LINK]

We look forward to receiving the revised manuscript by Dec 07 2022 11:59PM.   

Sincerely,

Callam Davidson, 

Associate Editor 

PLOS Medicine

plosmedicine.org

Comments from the Academic Editor:

I think the authors did a very good job in revising this and addressing the comments of all reviewers (except of the point on language). While the statistical reviewer is right about the limitations of only including English literature, this is an excellent review of the English literature. Furthermore including other language at this point would take a lot of time and is not really possible within the purview of a revision. From my reading of the literature, I don't expect substantial bias.

Requests from Editors:

Abstract Methods and Findings:

* Please include the study design (systematic review and meta-analysis).

* The sentence "Also, our study does not cover the entire care cascade as we did not measure gaps in pre-testing, pretreatment, and post-treatment outcomes (like loss-to-follow-up and TB recurrence)" should be relocated to the last sentence of the previous section (Abstract Methods and Findings).

Please include the decision not to include non-English language sources as a limitation in the Discussion.

Author Summary:

* Please reformat your Author Summary to comprise 2-3 single sentence bullet points for each of the three questions.

* Under the ‘What Did the Researchers Do and Find?’ question, please include the headline numbers from the study, such as the number of studies included and key findings. 

Lines 105-106: Please temper claims of primacy of results by stating, "to our knowledge" or something similar.

Please update your PRISMA checklist to use section and paragraph numbers, rather than line numbers (as these will not be present in the published version).

Please add the following statement, or similar, to the Methods: "This study is reported as per the Preferred Reporting Items for Systematic Reviews and Meta-Analyses (PRISMA) guideline (S1 Checklist)."

Line 137: I could not locate S1 Table in the Supporting Information.

Line 153: Figure 2 does not appear to relate to this statement. 

Please revise Figure 2 such that the word ‘Interventions’ does not break across two lines.

Table 2: Please define the abbreviation MDR in the figure legend.

Line 237: Please delete the duplicate [S6 Table] citation here.

In Figures 3A through 3F, please ensure that the X axis is identical for all figures to facilitate comparison.

The title of S9 Figure Panel A (TB Testing) has been cut off during creation of the figure.

Line 402: I could not locate Table 5.

S11 Figure appears to also duplicate Figures S9 and S10.

Line 825: Error in the numbering of the References here, please delete. 

Comments from Reviewers:

Reviewer #1: See attachment

Michael Dewey

Reviewer #4: I have no further comments

The new Tables are very helpful and clear

I stand corrected on the linkage to care definition

thank you

[LINK]

---

## [Editor Report · Decision Letter 3]

13 Dec 2022

Dear Dr Tang, 

On behalf of my colleagues and the Academic Editor, Dr Claudia Denkinger, I am pleased to inform you that we have agreed to publish your manuscript "Improving Cascade Outcomes for Active TB: A Global Systematic Review and Meta-Analysis of TB Interventions" (PMEDICINE-D-22-02722R3) in PLOS Medicine.

PRESS

Sincerely, 

Callam Davidson 

Associate Editor 

PLOS Medicine